# Statistical learning shapes pain perception and prediction independently of external cues

Jakub Onysk[1,2]*, Nicholas Gregory[1], Mia Whitefield[1], Maeghal Jain[1], Georgia Turner[1,3], Ben Seymour[4,5], Flavia Mancini[1]*

[1]Computational and Biological Learning Unit, Department of Engineering, University of Cambridge, Cambridge, United Kingdom; [2]Applied Computational Psychiatry Lab, Max Planck Centre for Computational Psychiatry and Ageing Research, Queen Square Institute of Neurology and Mental Health Neuroscience Department, Division of Psychiatry, University College London, London, United Kingdom; [3]MRC Cognition and Brain Sciences Unit, University of Cambridge, Cambridge, United Kingdom; [4]Wellcome Centre for Integrative Neuroimaging, John Radcliffe Hospital, Headington, Oxford, United Kingdom; [5]Center for Information and Neural Networks (CiNet), Osaka, Japan

*For correspondence:
jakub.onysk.22@ucl.ac.uk (JO);
fm456@cam.ac.uk (FM)

Competing interest: The authors declare that no competing interests exist.

**Abstract** The placebo and nocebo effects highlight the importance of expectations in modulating pain perception, but in everyday life we don't need an external source of information to form expectations about pain. The brain can learn to predict pain in a more fundamental way, simply by experiencing fluctuating, non-random streams of noxious inputs, and extracting their temporal regularities. This process is called statistical learning. Here, we address a key open question: does statistical learning modulate pain perception? We asked 27 participants to both rate and predict pain intensity levels in sequences of fluctuating heat pain. Using a computational approach, we show that probabilistic expectations and confidence were used to weigh pain perception and prediction. As such, this study goes beyond well-established conditioning paradigms associating non-pain cues with pain outcomes, and shows that statistical learning itself shapes pain experience. This finding opens a new path of research into the brain mechanisms of pain regulation, with relevance to chronic pain where it may be dysfunctional.

## eLife assessment

This study presents a **valuable** insight into a computational mechanism of pain perception. The evidence supporting the authors' claims is **compelling**. The work will be of interest to pain researchers working on computational models and cognitive mechanisms of pain in a Bayesian framework.

## Introduction

Clinical pain typically varies over time; in most pain states, the brain receives a stream of volatile and noisy noxious signals, which are also auto-correlated in time. The temporal structure of these signals is important, because the human brain has evolved the exceptional ability to extract regularities from streams of auto-correlated sensory signals, a process called statistical learning (*Dehaene et al., 2015*; *Schapiro and Turk-Browne, 2015*; *Fiser and Lengyel, 2019*; *Meyniel et al., 2016*; *Kourtzi and Welchman, 2019*; *Sherman et al., 2020*; *Turk-Browne et al., 2009*). In the context of pain, statistical

learning can allow the brain to predict future pain, which is crucial for orienting behaviour and maximising well-being (*Mancini et al., 2022*; *Mulders et al., 2023*). Statistical learning might also be fundamental to the ability of the nervous system to endogenously regulate pain. Indeed, statistical learning generates predictions about forthcoming pain. We already know that pain expectations can modulate pain levels by gating the reciprocal transmission of neural signals between the brain and spinal cord, as shown by previous work on placebo and nocebo effects (*Tracey, 2010*; *Tinnermann et al., 2017*; *Eippert et al., 2009*; *Geuter and Büchel, 2013*; *Fields, 2018*).

By using temporal sequences of noxious inputs, we have previously shown that the pain system supports the statistical learning of the basic rate of getting pain by engaging both somatosensory and supramodal cortical regions (*Mancini et al., 2022*). Specifically, both sensorimotor cortical regions and the ventral striatum encode probabilistic predictions about pain intensity, which are updated as a function of learning by engaging parietal and prefrontal regions. According to a Bayesian inference framework, both the predictive inference and its confidence should, in principle, modulate the neural response to noxious inputs and affect perception, as a function of learning. In support of this conjecture, there is evidence that the confidence of probabilistic pain predictions modulates the cortical response to pain (*Mulders et al., 2023*). The relationship is inverse: the lower the confidence, the higher is the early cortical response to noxious inputs (and vice versa), as measured by EEG. This is expected based on Bayesian inference theory: when confidence is low, the brain relies less on his prior beliefs and more on sensory evidence to respond to the input. Bayesian inference theory also predicts that prior expectations and their confidence scale perception (*Knill and Richards, 1996*). Thus, we hypothesise that the predictions generated by learning the statistics of noxious inputs in dynamically evolving sequences of stimuli modulate the perception of forthcoming inputs.

Previously, it was found that pain perception is strongly influenced by probabilistic expectations as defined by a cue that predicts high or low pain (*Jepma et al., 2018*). In contrast to such cue paradigm, the primary aim of our experiment was to determine whether the expectations participants hold about the sequence itself inform their perceptual beliefs about the intensity of the stimuli. To that end, we recruited 27 healthy participants to complete a psycho-physical experiment where we delivered four different, 80-trial-long sequences of evolving thermal stimuli, with four levels of temporal regularity. On each trial, a 2 s thermal stimulus was applied, following which participants were asked to either rate their perception of the intensity (*Figure 1A*) or to predict the intensity of the next stimulus in the sequence (*Figure 1B*). Participants also reported their response confidence.

We contrasted four models of statistical learning, which varied according to the inference strategy used (i.e. optimal Bayesian inference or a heuristic) and the role of expectations on perception. All models used confidence ratings to weigh the inference. We anticipate that probabilistic learning weighted by confidence and expectations modulates pain perception. This provides behavioural evidence for a link between learning and endogenous pain regulation. One reason why this is important is that it might help understand individual differences in the ability to endogenously regulate pain. This is particularly relevant for chronic pain, given that endogenous pain regulation can be dysfunctional in several chronic pain conditions (*Bushnell et al., 2013*; *Bruehl et al., 1999*; *Yarnitsky, 2015*; *King et al., 2009*; *Bannister and Dickenson, 2017*), even before chronic pain develops (*Tracey, 2016*). Although there is ample evidence for changes in the functional anatomy and connectivity of endogenous pain modulatory systems in chronic pain, their computational mechanisms are poorly understood.

## Results

### Model-naive performance

Prior to modelling, we first checked whether participant's performance in the task was affected by the level of temporal regularity, i.e., the sequence condition. We varied the level of volatility and stochasticity across blocks (i.e. conditions), whilst we fixed their overall level within each block; the level of volatility was defined by the number of trials until the mean intensity level changes. The stochasticity is the additional noise that is added on each trial to the underlying mean, often referred to as the observation noise. The changes were often subtle and participants were not informed when they happened. We set two levels (low/high) of each type of uncertainty, achieving a 2×2 factorial design, with the order of conditions randomised across participants. A set of four example sequences of thermal

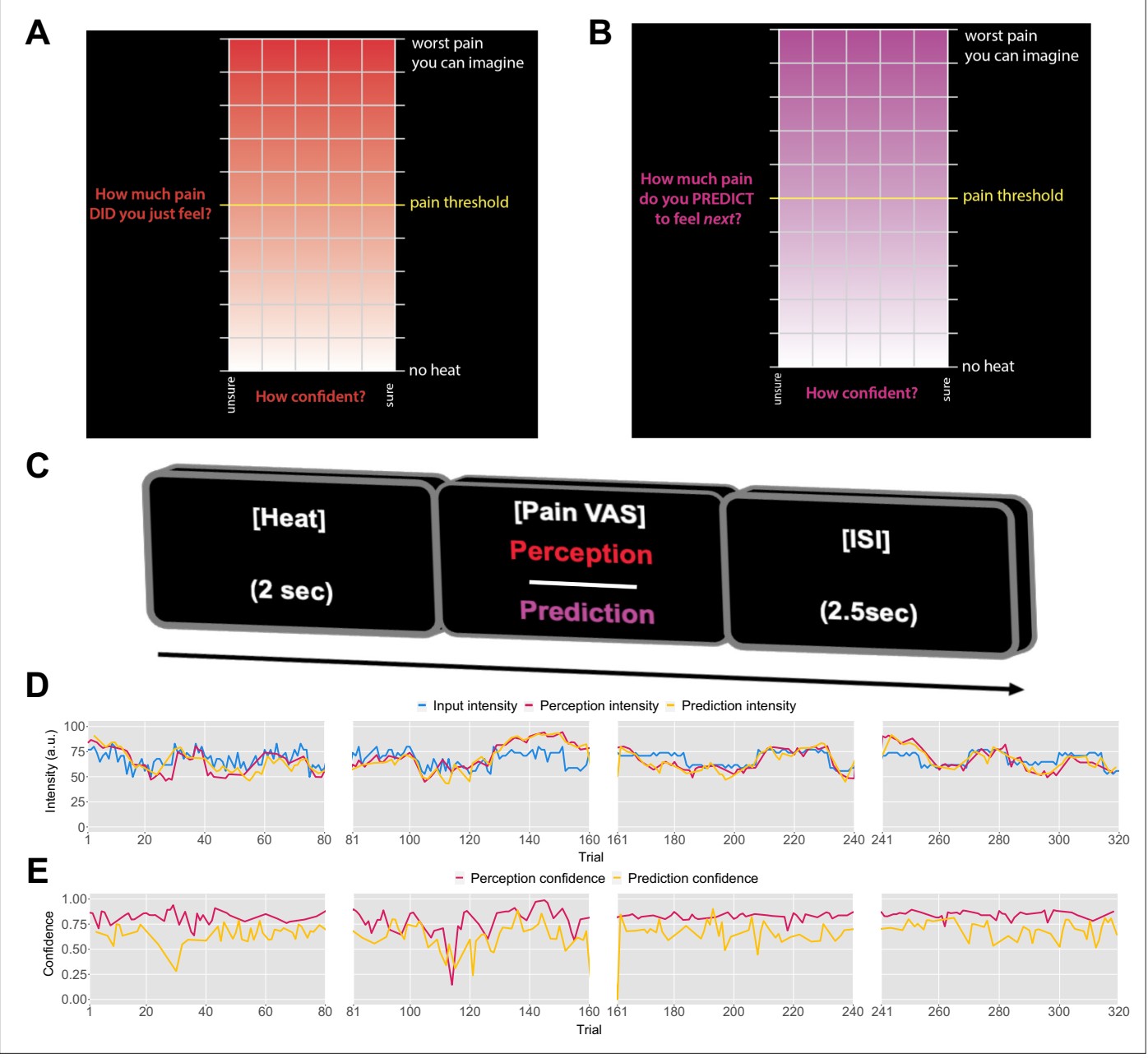

**Figure 1.** Task design. On each trial, each participant received a thermal stimulus lasting 2s from a sequence of intensities. This was followed by a perception (**A**) or a prediction (**B**) input screen, where the *y*-axis indicates the level of perceived/predicted intensity (0–100) centred around participant's pain threshold, and the *x*-axis indicates the level of confidence in one's perception (0–1). The inter-stimulus interval (ISI; black screen) lasted 2.5s (trial example in **C**). (**D**) Example intensity sequences are plotted in green, participant's perception and prediction responses are in red and black, respectively. (**E**) Participant's confidence rating for perception (red) and prediction (black) trials.

intensities delivered to one of the participants can be found in *Figure 1C*, alongside their ratings of perception and predictions. Additionally, example confidence ratings for each type of response are plotted in *Figure 1D*. *Appendix 1—figures 2 and 3* show the plots of each participant's responses superimposed onto the sequences of noxious inputs.

As a measure of performance, we calculated the root mean square error (RMSE) of participants responses (ratings and predictions) compared to the normative noxious input for each condition as in *Figure 2* (see also Materials and methods). The lower the RMSE, the more accurate participants' responses are. Performance in different conditions was analysed with a repeated measures ANOVA,

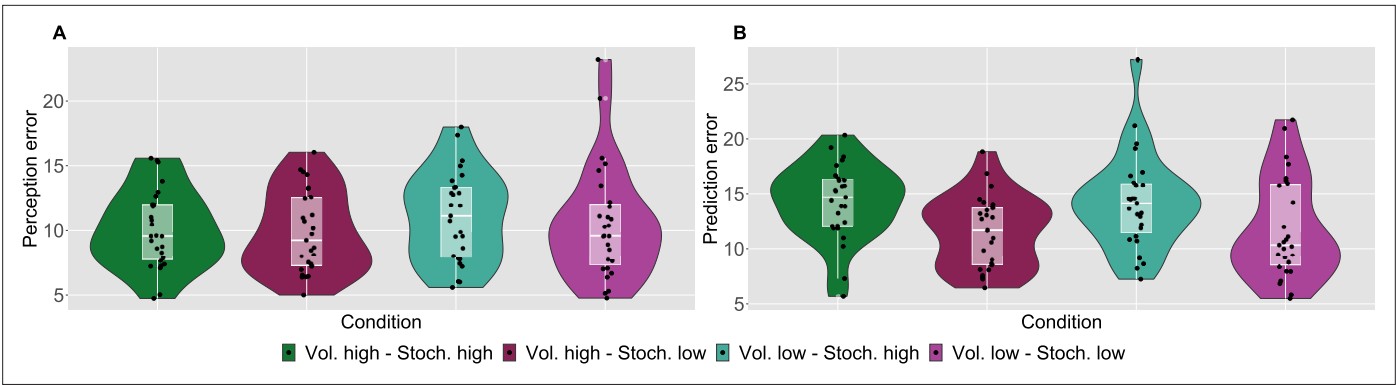

**Figure 2.** Participant's model-naive performance in the task. Violin plots of participant root mean square error (RMSE) for each condition for **A:** rating and **B:** prediction responses as compared with the input. Lower and upper hinges correspond to the first and third quartiles of partipants' errors (the upper/lower whisker extends from the hinge to the largest/smallest value no further than 1.5 * "Interquartile range" from the hinge); the line in the box corresponds to the median. Each condition has N=27 particpants.

whose results are reported in full in *Appendix 1—table 1*. Although volatility did not affect rating accuracy $(F(1, 26) = 0.96, \text{p} = 0.336, \eta_p^2 = 0.036)$, we found a two-way interaction between the level of stochasticity of the sequence (low, high) and the type of rating provided (perceived intensity vs. prediction) $(F(1, 26) = 29.842, \text{p} < 0.001, \eta_p^2 = 0.534)$. We followed up this interaction in post hoc comparisons, as reported in *Appendix 1—table 2*. The performance score differences between all the pairs of stochasticity and response type interactions were significant, apart from the perception ratings in the stochastic environment as compared with perception and prediction performance in the low stochastic setting. Intuitively, the RMSE score analysis revealed an overall trend of participants performing worse on the prediction task, in particular when the level of stochasticity is high.

## Modelling strategy

Our models were selected a priori, following the modelling strategy from *Jepma et al., 2018*, and hence considered the same set of core models for clear extension of the analysis to our non-cue paradigm. The key question for us was whether expectations were used to weigh the behavioural estimates during sequence learning. Therefore, we compared Bayesian and non-Bayesian models of sequential learning that weighted their ratings based on prior expectations versus two corresponding models that assumed perfect perception (i.e. not weighted by prior beliefs). As a baseline, we included a random response model (please see Materials and methods for a formal treatment of the computational models).

According to an optimal Bayesian inference strategy, on each trial, participants update their beliefs about the feature of interest (thermal stimuli) based on probabilistic inference, maintaining a full posterior distribution over its values (*Jepma et al., 2018*; *Särkkä, 2013*). Operating within a Bayesian paradigm, participants are assumed to track and, following new information, update both the mean of the sequence of interest and the uncertainty around it (*Hoskin et al., 2019*). In most cases, such inference makes an assumption about environmental dynamics. For example, a common assumption is that the underlying mean (a hidden/latent state) evolves linearly according to a Gaussian random walk, with the rate of this evolution defined by the the variance of this Gaussian walk (volatility). The observed value is then drawn from another Gaussian with that mean, which has some observation noise (stochasticity). In this case, the observer can infer the latent states through the process of Bayesian filtering (*Särkkä, 2013*), using the Kalman filter (KF) algorithm (*Kalman, 1960*; *Figure 3B*).

Sequence learning can also be captured by a heuristic to the Bayesian inference, i.e., a simple reinforcement learning (RL) rule. Here, participants maintain and update a point estimate of the expected value of the sequence in an adaptive manner, within a non-stationary environment. RL explicitly involves correcting the tracked mean of the sequence proportionally to a trial-by-trial prediction error - a difference between the expected and actual value of the sequence (*Sutton and Barto, 2018*; *Figure 3A*). Importantly, RL agents do not assume any specific dynamics of the environment and hence are considered model-free.

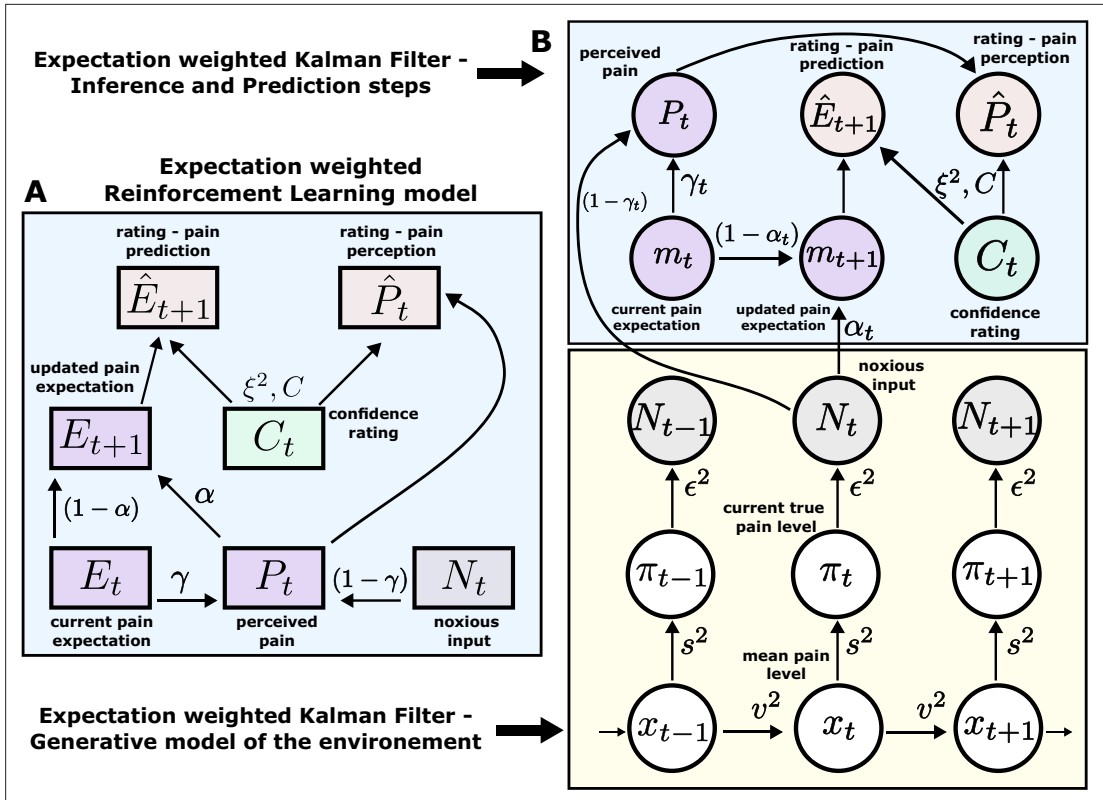

**Figure 3.** Expectation weighted models. Computational models used in the main analysis to capture participants' pain perception ($P_t$) and prediction ($E_{t+1}$) ratings. Both types of ratings are affected by confidence rating ($C_t$) on each trial. (**A**) In the reinforcement learning model, participant's pain perception ($P_t$) is taken to be weighted sum of the current noxious input ($N_t$) and their current pain expectation ($E_t$). Following the noxious input, participant updates their pain expectation ($E_{t+1}$). (**B**) In the Kalman filter model, a generative model of the environment is assumed (yellow background) - where the mean pain level ($x_t$) evolves according to a Gaussian random walk (volatility $v^2$). The true pain level on each trial ($\pi_t$) is then drawn from a Gaussian (stochasticity $s^2$). Lastly, the noxious input, $N_t$, is assumed an imperfect indicator of the true pain level (subjective noise $\epsilon^2$). Inference and prediction steps are depicted in a blue box. Participant's perceived pain is a weighted sum of expectation about the pain level ($m_t$) and current noxious input ($N_t$). Following each observation, $N_t$, participant updates their expectation about the pain level ($m_{t+1}$).

Both models perform a form of error correction about the underlying sequence. The rate at which this occurs is captured by the learning rate $\alpha \in [0, 1]$ element. The higher the learning rate, the faster participants update their beliefs about the sequence after each observation. For the RL model, the learning rate $\alpha$ is a free parameter that is constant across the trials. On the other hand, the learning rate in the KF model $\alpha_t$ (known as the Kalman gain) is calculated on every trial. It depends on participants' trial-wise belief uncertainty as well as their overall estimation of the inherent noise in the environment (stochasticity, $s$). The belief uncertainty is updated after each observation and depends on participants' sense of volatility ($v$) and stochasticity ($s$) in the environment.

Crucially, we also used participants' trial-by-trial confidence ratings to measure to what extent confidence plays a role in learning. This is captured by the confidence scaling factor $C$, which defines the extent to which confidence affects response (un-)certainty. Intuitively, the higher the confidence scaling factor $C$, the less important role confidence plays in participant's response. With relatively low values of $C$, when the confidence is low, participants' responses are more noisy, i.e., less certain. We demonstrate this in *Figure 4* by plotting hypothetical responses (A–F) and the effect on the noise scaling (G–L) as a function of $C$ and confidence ratings.

To evaluate the effect of expectation on perceived intensity (on top of statistical learning modulating perception), we expanded the standard RL and KF models by adding a perceptual weighting element, $\gamma \in [0, 1]$ (similarly to *Jepma et al., 2018*). Essentially, $\gamma$ governs how much each participant relies on the normative input on each trial, and how much their expectation of the input influences their reported perception - i.e., they take a weighted average of the two. The higher the $\gamma$, the bigger the impact of the expectation on perception. Again, in the case of the RL model (eRL - expectation

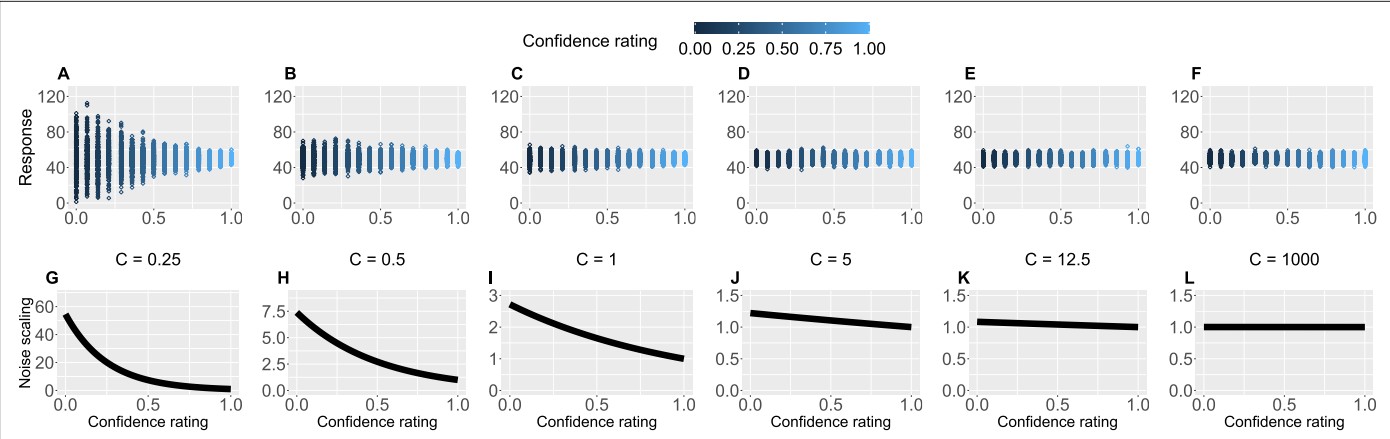

**Figure 4.** Confidence scaling factor demonstration. (**A–F**) For a range of values of the confidence scaling factor $C$, we simulated a set of typical responses a participant would make for various levels of confidence ratings. The belief about the mean of the sequence is set at 50, while the response noise at 10. The confidence scaling factor $C$ effectively scales the response noise, adding or reducing response uncertainty. (**G–L**) The effect of different levels of parameter $C$ on noise scaling. As $C$ increases the effect of confidence is diminished.

weighted RL), $\gamma$ is a free parameter that is constant across trials, while in the KF model (eKF - expectation weighted KF), $\gamma_t$ is calculated on every trial and depends on: (1) the participants' trial-wise belief uncertainty, (2) their overall estimation of the inherent noise in the environment (stochasticity, $s$), and (3) the participant's subjective uncertainty about the level of intensity, $\epsilon$.

Thus, in total we tested five models: RL and KF (perception not weighted by expectations), eRL and eKF (perception weighted by expectations), and a baseline random model. We then proceeded to fit these five computational models to participants' responses. For parameter estimation, we used hierarchical Bayesian methods, where we obtained group- and individual-level estimates for each model parameter (see Materials and methods).

## Modelling results

We fit each model for each condition sequence. Example trial-by-trial model prediction plots from one participant can be found in *Appendix 1—figure 4*. To establish which of the models fitted the data best, we ran model comparison analysis based on the difference in expected log point-wise predictive density (ELPD) between models. The models are ranked according to the ELPD (with the largest providing the best fit). The ratio between the ELPD difference and the standard error around it provides a significance test proxy through the sigma effect. We considered at least a 2 sigma effect as indication of a significant difference. In each condition, the expectation weighted models (eKF and eRL) provided better fit than models without this element (KF and RL; approximately 2–4 sigma difference, as reported in *Figure 5A–D*) and *Appendix 1—table 5*. This suggests that regardless of the levels of volatility and stochasticity, participants still weigh perception of the stimuli with their expectation. In particular, we found that the expectation weighted KF model offered a better fit than the eRL, although in conditions of high stochasticity this difference was short of significance against the eRL model. This suggests that in learning about temporal regularities in the sequences of thermal stimuli, participants' expectations modulate the perception of the stimulus. Moreover, this process was best captured by a model that updates the observer's belief about the mean and the uncertainty of the sequence in a Bayesian manner.

We also found that as the confidence in the response decreases, the response uncertainty is scaled linearly with a negative slope ranging between 0.112 and 0.276 across conditions (*Figure 6*), confirming the intuition that less confidence leads to bigger uncertainty.

As an additional check, for each participant, condition and response type (perception and prediction), we plotted participants' ratings against model predicted ratings and calculated a grand mean correlation in *Appendix 1—figure 5*.

Next, we checked whether the parameters of the the winning eKF model differed across different sequence conditions. Given that volatility was fixed within condition, we treated it as a single-context

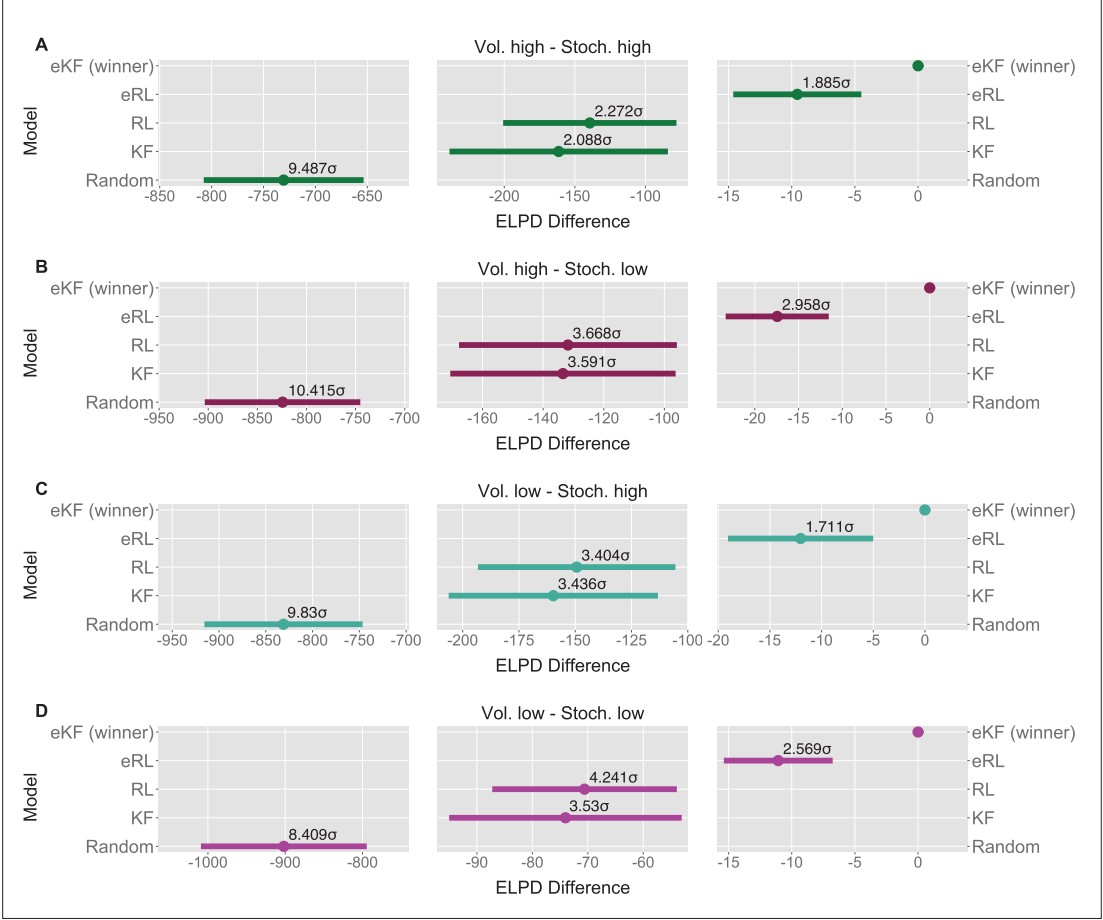

**Figure 5.** Model comparison for each sequence condition (**A–D**). The dots indicate the expected log point-wise predictive density (ELPD) difference between the winning model (eKF - expectation weighted Kalman filter) and every other model. The line indicates the standard error (SE) of the difference. The non-winning models' ELPD differences are annotated with the ratio between the ELPD difference and SE indicating the sigma effect, a significance heuristic.

scenario from the point of view of modelling (*Heald et al., 2023*), and we did not interpret its effect on the learning rate (*Piray and Daw, 2021*). There were no differences for the group-level parameters; i.e., we did not detect significant differences between conditions in a hypothetical healthy participant group as generalised from our population of participants (*Appendix 1—figure 12*).

However, we found some differences at the individual level of parameters (i.e. within our specific population of recruited participants), which we detected by performing repeated measures ANOVAs

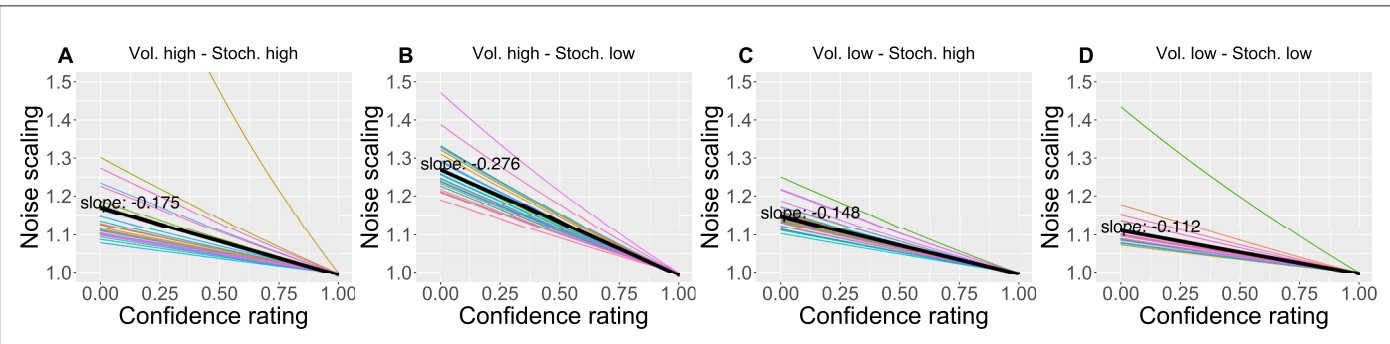

**Figure 6.** The effect of the confidence scaling factor on noise scaling for each condition. (**A–D**) Each coloured line corresponds to one participant, with the black line indicating the mean across all participants. The mean slope for each condition is annotated.

(see *Appendix 1—figure 13* for visualisation). The stochasticity parameter $s$ was affected by the interaction between the levels of stochasticity and volatility ($F(1, 26) = 35.108$, p $< 0.001$, $\eta_p^2 = 0.575$), and was higher in highly stochastic and volatile conditions as compared to conditions where either volatility ($t = 7.735$, p$_{bonf} < 0.001$), stochasticity ($t = 9.396$, p$_{bonf} < 0.001$), or both were low ($t = 8.826$, p$_{bonf} < 0.001$). This suggests that, while participants' performance was generally worse in highly stochastic environments, participants seem to have attributed this to only one source - stochasticity ($s$), regardless of the source of higher uncertainty in the sequence (stochasticity or volatility).

The response noise $\xi$ was modulated by the level of volatility ($F(1, 26) = 5.079$, p $= 0.033$, $\eta_p^2 = 0.163$), where it was smaller in highly volatile conditions. Moreover, we detected a significant interaction between volatility and stochasticity on the confidence scaling factor $C$ ($F(1, 26) = 81.258$, p $< 0.001$, $\eta_p^2 = 0.758$), where the values $C$ were overall lower when either volatility ($t = -11.570$, p$_{bonf} < 0.001$), stochasticity ($t = -6.165$, p$_{bonf} < 0.001$), or both ($t = -4.575$, p$_{bonf} < 0.001$) were high as compared to the conditions where both levels of noise were low. This indicates there may have been some trade-off between $\xi$ and $C$, as lower values of $C$ introduce additional uncertainty when participant's confidence is low.

Lastly, we found the initial uncertainty belief $w_0$ was affected by the interaction between volatility and stochasticity ($F(1, 26) = 5275.367$, p $< 0.001$, $\eta_p^2 = 0.995$) without a consistent pattern. All the other effects were not significant.

In summary, we formalised the process behind pain perception and prediction in noxious time-series within the framework of sequential learning, where the best description of participants' statistical learning was captured through Bayesian filtering, in particular using a confidence weighted KF model. Most importantly, we discovered that, in addition to weighing their responses with confidence, participants used their expectations about stimulus intensity levels to form a judgement as to what they perceived. This mechanism was present across various levels of uncertainty that defined the sequences (volatility and stochasticity).

## Discussion

Statistical learning allows the brain to extract regularities from streams of sensory inputs and is central to perception and cognitive function. Despite its fundamental role, it has often been overlooked in the field of pain research. Yet, chronic pain appears to fluctuate over time. For instance *Mayr et al., 2022*; *Baliki et al., 2012*; *Foss et al., 2006*, reported that chronic back pain ratings vary periodically, over several seconds-minutes and in absence of movements. This temporal aspect of pain is important because periodic temporal structures are easy to learn for the brain (*Dehaene et al., 2015*; *Mancini et al., 2022*). If the temporal evolution of pain is learned, it can be used by the brain to regulate its responses to forthcoming pain, effectively shaping how much pain it experiences. Indeed, we show that healthy participants extract temporal regularities from sequences of noxious stimuli and use this probabilistic knowledge to form confidence weighted judgements and predictions about the level of pain intensity they experience in the sequence. We formalised our results within a Bayesian inference framework, where the belief about the level of pain intensity is updated on each trial according to the amount of uncertainty participants ascribe to the stimuli and the environment. Importantly, their perception and prediction of pain were influenced by the expected level of intensity that participants held about the sequence before responding. When varying different levels of inherent uncertainty in the sequences of stimuli (stochasticity and volatility), the expectation and confidence weighted models fitted the data better than models weighted for confidence but not for expectations (*Figure 5A–D*). The expectation weighted Bayesian (eKF) model offered a better fit than the expectation weighted, model-free RL model, although in conditions of high stochasticity this difference was short of significance. Overall, this suggests that participants' expectations play a significant role in the perception of sequences of noxious stimuli.

### Statistical inference and learning in pain sequences

The first main contribution of our work is towards the understanding of the phenomenon of statistical learning in the context of pain. Statistical learning is an important function that the brain employs across the lifespan, with relevance to perception, cognition, and learning (*Sherman et al., 2020*). The large majority of past research on statistical learning focused on visual and auditory perception (*Dehaene et al., 2015*; *Fiser and Lengyel, 2019*; *Meyniel et al., 2016*; *Meyniel and Dehaene,*

*2017*), with the nociceptive system receiving relatively little attention (*Tabor et al., 2017*). Recently, we showed that the human brain can learn to predict a sequence of two pain levels (low and high) in a manner consistent with optimal Bayesian inference, by engaging sensorimotor regions, parietal, premotor regions, and dorsal striatum (*Mancini et al., 2022*). We also found that the confidence of these probabilistic inferences modulates the cortical response to pain, as expected by hierarchical Bayesian inference theory (*Mulders et al., 2023*). Here, we tested sequences with a much larger range of stimulus intensities to elucidate the effect of statistical learning and expectations on pain perception. As predicted by hierarchical Bayesian inference theory, we find that the pain intensity judgements are scaled by both expectations and confidence.

Hence, our work highlights the inferential nature of the nociceptive system (*Jepma et al., 2018*; *Tabor et al., 2017*; *Fardo et al., 2017*; *Seymour and Mancini, 2020*; *Büchel et al., 2014*), where in addition to the sheer input received by the nociceptors, there is a wealth of a priori knowledge and beliefs the agent holds about themselves and the environment that need to be integrated to form a judgement about pain (*Yoshida et al., 2013*; *Anchisi and Zanon, 2015*; *Wiech, 2016*; *Tabor and Burr, 2019*). This has an immediate significance for the real world, where weights need to be assigned to prior beliefs and/or stimuli to successfully protect the organism from further damage, but only to an extent to which it is beneficial.

Secondly, our results regarding the effect of expectation on pain perception relate to a much larger literature on this topic. The prime example would be placebo analgesia (i.e. the expectation of pain relief decreasing pain perception) and nocebo hyperalgesia (i.e. the expectation of high level of pain increasing its perception; *Tracey, 2010*; *Büchel et al., 2014*; *Colloca et al., 2008*; *Blasini et al., 2017*). Recent work attempted to capture such expectancy effects within the Bayesian inference framework. For example, *Hoskin et al., 2019*, showed that in addition to expectation influencing perceived pain in general, higher level of uncertainty around that expectation attenuated its effect on perception. Similarly, *Hird et al., 2019*, demonstrated that when the discrepancy between the expectation and outcome (prediction error) is unusually large, the role of expectation is significantly reduced and so the placebo and nocebo effects are not that strong. An unusually large prediction error could be thought of as contributing to increased uncertainty about the stimuli, which mirrors the results from *Hoskin et al., 2019* Bayesian framework. Nevertheless, the types of stimuli used in the above studies (i.e. noxious stimuli cued by non-noxious stimuli) differed from the more ecologically valid sequences of pain that are reported by chronic pain patients (*Mayr et al., 2022*), as we indicated above. Furthermore, *Jepma et al., 2018*, used a conditioning paradigm and also found that expectations influence both perception and learning, in a self-reinforcing loop. Our work has followed a similar modelling strategy to *Jepma et al., 2018*, but it goes beyond simple conditioning schedules or sequences of two-level discrete painful stimuli, showing expectancy effects even when the intensities are allowed to vary across a wider range of values and according to more complex statistical temporal structures. Additionally, given the reported role of confidence in the perception of pain (*Mulders et al., 2023*; *Brown et al., 2008*), we draw a more complete picture by including participants' confidence ratings in our modelling analysis.

Future studies would need to determine whether statistical learning and its effect on pain is altered in chronic pain conditions. This is important because statistical learning could, in principle, influence how a pain state evolves. Once a pain state is initiated, how an individual learns and anticipates the fluctuating pain signals may contribute to determine how well it can be regulated by the nervous system, thus affecting the severity and recurrence of pain flares. This, in turn, would affect whether aversive associations with the instigating stimulus are extinguished or reinforced (*Seymour and Mancini, 2020*). In chronic pain, dysfunctional learning may promote the amplification and mainte-nance of pain signals, contributing to the reinforcement of aversive associations with incident stimuli, as well as the persistence of pain (*Seymour, 2019*; *Baliki and Apkarian, 2015*; *Vlaeyen et al., 2016*).

Our paper comes with open tools, which can be adapted in future studies on statistical learning in chronic pain. The key advantage of taking an hypothesis-driven, computational-neuroscience approach to quantify learning is that it allows to go beyond symptoms-mapping, identifying the quan-tifiable computational principles that mediate the link between symptoms and neural function.

In summary, we show that statistical expectations and confidence scale the judgement of pain in sequences of noxious stimuli as predicted by hierarchical Bayesian inference theory, opening a new avenue of research on the role of learning in pain.

# Materials and methods

## Participants

Thirty-three (18 female) healthy adult participants were recruited for the experiment. The mean age of participants was 22.4±2.7 years of age (range: 18–35). Participants had no chronic condition and no infectious illnesses, as well as no skin conditions (e.g. eczema) at the site of stimulus delivery. Moreover, we only recruited participants that had not taken any anti-anxiety, anti-depressive medication, nor any illicit substances, alcohol and pain medication (including NSAIDs such as ibuprofen and paracetamol) in the 24 hr prior to the experiment. All participants gave informed written consent to take part in the study, which was approved by the local ethics committee (Department of Engineering, University of Cambridge Ethics Committee).

## Protocol

The experimental room's temperature was maintained between 20°C and 23°C. Upon entry, an infrared thermometer was used to ensure participants' temperature was above 36°C at the forehead and forearm of the non-dominant hand, to account for the known effects of temperature on pain perception (*Strigo et al., 2000*). A series of slideshows were presented, which explained the premise of the experiment and demonstrated what the participant would be asked to carry out. Throughout this presentation, questions were asked to ensure participants understood the task. Participants were given multiple opportunities to ask questions throughout the presentation.

We used the Medoc Advanced Thermosensory Stimulator 2 (TSA2) (*Medoc Advanced Medical Systems, 2022*) to deliver thermal stimuli using the CHEPS thermode. The CHEPS thermode allowed for rapid cooling (40°C/s) and heating (70°C/s) so transitions between the baseline and stimuli temperatures were minimal. The TSA2 was controlled externally, via Matlab (Mathworks).

We then established the pain threshold, using the method of limits (*Lue and Shih, 2017*), in order to centre the range of temperature intensities used in the experiment. Each participant was provided with stimuli of increasing temperature, starting from 40°C going up in 0.5°C increments, using an inter-stimulus interval (ISI) of 2.5 s and a 2 s duration. The participant was asked to indicate when the stimuli went from warm to painful - this temperature was noted and the stimuli ended. The procedure was repeated three times, and the average was used as an estimate of the pain threshold.

During the experiment, four sequences of thermal stimuli were delivered. Due to the phenomenon of offset analgesia, where decreases in tonic pain result in a proportionally larger decrease in perceived pain (*Hermans et al., 2016*), we chose phasic stimuli, with a duration of 2 s and an ISI 2.5 s. In order to account for individual differences, the temperatures which the levels refer to are based upon the participants' threshold. The median intensity level was defined as threshold, giving a max temperature of 3°C above threshold, which was found to be acceptable by participants. Before the start of the experiment each participant was provided with the highest temperature stimuli that could be presented, given their measured threshold, to ensure they were comfortable with this. Two participants found the stimulus too painful - the temperature range was lowered by 1°C and this was found to be acceptable.

After every trial of each sequence, the participant was asked for either their perception of the previous stimulus or their prediction for the next stimulus through a 2D VAS (*Figure 1B*), presented using PsychToolBox-3 (*Kleiner, 2007*). The y-axis encodes the intensity of the stimulus either perceived or predicted, ranging from 0 (no heat detected/predicted) to 100 (worst pain imaginable perceived/predicted); on this scale, 50 represents the pain threshold. This was done as a given sequence was centred around the threshold. The x-axis encodes confidence in either perception or prediction, ranging from 0 - completely uncertain ('unsure') - to 1 - complete confidence in the rating ('sure'). Differing background colours were chosen to ensure participants were aware of what was being asked, and throughout the experiment participants were reminded to take care in answering the right question. The mouse movement was limited to be inside of the coloured box, which defined the area of participants' input. At the beginning of each input screen, the mouse location was uniformly randomised within the input box.

The sequence of response types was randomised so as to retain 40 prediction and 40 perception ratings for each of the four sequence conditions. For an 80-trial-long sequence, this gave 80 participant responses. Each sequence condition was separated by a 5 min break, during which the thermode's probe was slightly moved around the area of skin on the forearm to reduce sensitisation

(i.e. a gradual increase in perceived intensity with repetitive noxious stimuli) (*Hollins et al., 2011*). In the middle of each sequence, there was a 3 min break. During the ISI, the temperature returned to a baseline of 38°C. One participant was unable to complete the sequence as their threshold was too low, and data from four participants was lost due to Medoc software issues (the remote control failed and the data of two out of four sessions were not saved). We excluded one participant's whose ratings/predictions were inversely proportional to the noxious input. Thus, we analysed data from 27 participants.

## Generative process of the painful sequences

We manipulated two sources of uncertainty in the sequence: the stochasticity (*s*) of the observation and the volatility (*v*) of the underlying sequence (*Piray and Daw, 2021*). Sequences were defined by two levels (high or low) of stochasticity and volatility, resulting in four different sequences conditions - creating a 2×2 factorial design. Each sequence was defined as a series of chunks, where the intensity for trial $t$, $i_t$, was sampled from $N\left(I, \sigma^2\right)$, where $\sigma^2$ indicates the level of stochasticity ($\sigma^2 = 1.75$ for high level of stochasticity, $\sigma^2 = 0.25$ for low level of stochasticity). The mean of the chunk, $I$, was drawn from $U\left(3.5, 10.5\right)$. To ensure a noticeable difference in chunk intensity to the participant, concurrent chunk means were constrained to be at least two intensity levels different. Volatility was implemented by defining the length, or number of trials, of a chunk ($l$) drawn from $U\left(L - a, L + a\right)$, where $L$ is the mean o the chunk length ($L = 15$ for high volatility level, $L = 25$ for low volatility level). A jitter, $a$, was added around the mean to ensure the transition from one chunk to the next was not consistent or predictable. For both high and low volatility conditions, we set $a = 3$. Sampled values were then discretised, where any intensities outside the valid intensity range $\left[1, 13\right]$ were discarded and re-sampled resulting in an 80-trial-long sequence for each condition. The mean of each sequence was centred around intensity level 7, i.e., the participants threshold. So defined, six sets of four sequences were sampled. Each participant received one set, with a randomised sequence order. See an example sequence (after subject-specific linear transformation) and one participant's responses (including confidence ratings) in *Figure 1C and D*.

## Data pre-processing

Since the intensity values of the noxious input were discretised between 1 and 13, while the participant's responses (perception and prediction) were given on a 0–100 scale, we applied a linear transformation of the input to map its values onto a common 0–100 range. For each participant, for a set of inputs at perception trials from the concatenated sequence (separate sequence conditions in the order as presented), we fit a linear least squares regression using Python's scipy.stats.linregress function. On rare occasions, when the transformed input was negative, we refit the line using Python's nonlinear least squares function scipy.optimize.curve_fit, constraining the intercept above 0 (*Virtanen et al., 2020*). We then extracted each participant's optimised slope and intercept and applied the transformation to both the concatenated and condition-specific sequence of inputs. So transformed, the sequences were then used in all the analyses. Plots of each participant transformation can be found in *Appendix 1—figure 1*. We superimposed participant's responses onto the noxious input condition sequences in *Appendix 1—figure 2*.

To capture participant's model-naive performance in the task, both for the concatenated and condition-specific sequence, we calculated RMSE of each participant's perception (*Equation 1*) and prediction (*Equation 2*) responses as compared to the input. The lower the RMSE, the higher the response accuracy.

$$RMSE_P = \sqrt{\frac{\sum_{t=1}^{T_P} \left(Y_t - \hat{P}_t\right)}{T_P}} \tag{1}$$

$$RMSE_E = \sqrt{\frac{\sum_{t=1}^{T_E} \left(y_{t+1} - \hat{E}_{t+1}\right)}{T_E}} \tag{2}$$

where $T_P$ is the number of perception trials, $\hat{P}_t$ is participant's perception response to the stimulus $y_t$ at trial $t$, $T_E$ is the number of prediction trials, and $\hat{E}_{t+1}$ is participant's prediction of the next stimulus intensity $y_{t+1}$ at trial $t + 1$.

## Models

### Reinforcement learning

#### RL

In RL models, learning is driven by discrepancies between the estimate of the expected value and observed values. Before any learning begins, at trial $t = 1$, participants have an initial expectation, $E_1 = E_0$, which is a free parameter that we estimate.

On each trial, participants receive a thermal input $N_t$. We then calculate the prediction error $\delta_t$, defined as the difference between the expectation $E_t$ and the input $N_t$ (**Equation 3**).

$$\delta_t = N_t - E_t \tag{3}$$

Participant is then assumed to update their expectation of the stimulus on the next trial as in **Equation 4**:

$$E_{t+1} = E_t + \alpha\delta_t \tag{4}$$

where $\alpha$ is the learning rate (free parameter), which governs how fast participants assimilate new information to update their belief.

On trials when participants rate their perceived intensity, we assume no effects on their perception other than confidence rating $c_t$ and response noise, so participants' perception response $P_t$ is drawn from a Gaussian distribution, with the mean $P_t = N_t$ and a confidence-scaled response noise $\xi$ (free parameter), as in **Equation 5**:

$$\hat{P}_t \sim \mathcal{N}\left(P_t, \xi^2 exp\left\{C^{-1}\left(1 - c_t\right)\right\}^2\right) \tag{5}$$

where $C$ is the confidence scaling factor (free parameter), which defines the extent to which confidence affects response uncertainty. Please see **Figure 4** for an intuition behind confidence scaling.

On trials when participants are asked to predict the intensity of the next thermal stimulus, we use the updated expectation $E_{t+1}$ to model participants' prediction response $E_{t+1}$. This is similarly affected by confidence rating and response noise and is defined as in **Equation 6**.

$$\hat{E}_{t+1} \sim \mathcal{N}\left(E_{t+1}, \xi^2 exp\left\{C^{-1}\left(1 - c_t\right)\right\}^2\right) \tag{6}$$

To recap, the RL model has four free parameters: the learning rate $\alpha$, response noise $\xi$, the initial expectation $E_0$, and the confidence scaling factor $C$.

#### eRL

Additionally, where we investigate the effects of expectation on the perception of pain (**Jepma et al., 2018**), we included an element that allows us to express the perception as a weighted sum of the input and expectation (**Equation 7**):

$$P_t = \left(1 - \gamma\right) N_t + \gamma E_t \tag{7}$$

where $\gamma \in \left[0, 1\right]$ (free parameter) captures how much participants rely on the normative thermal input vs. their expectation. When $\gamma = 0$, the expectation plays no role and the model simplifies to that of the standard RL above. In total, the eRL model has five free parameters, with the other equations the same as in the RL model, with the exception of the prediction error, which now relies on the expectation weighted pain perception $P_t$ (**Equation 8**).

$$\delta_t = P_t - E_t \tag{8}$$

## Kalman filter

### KF

To capture sequential learning in a Bayesian manner, we used the KF model (*Jepma et al., 2018*; *Särkkä, 2013*; *Kalman, 1960*). KF assumes a generative model of the environment where the latent state on trial $t$, $x_t$ (the mean of the sequences in the experiment), evolves according to a Gaussian random walk with a fixed drift rate, $v$ (volatility), as in *Equation 9*.

$$x_t \sim \mathcal{N}\left(x_{t-1}, v^2\right) \tag{9}$$

The observation on trial $t$, $N_t$, is then drawn from a Gaussian (*Equation 10*) with a fixed variance, which represents the observation uncertainty $s$ (stochasticity).

$$N_t \sim \mathcal{N}\left(x_t, s^2\right) \tag{10}$$

As such the KF assumes stable dynamics since the generative process has fixed volatility and stochasticity.

For ease of explanation, we refer to the thermal input at each trial as $N_t$, we also use the $N_{1:t}$ notation, which refers to a sequence of observations up to and including trial $t$. The model allows to obtain posterior beliefs about the latent state $x_t$ given the observations. This is done by tracking an internal estimate of the mean $m_t$ and the uncertainty, $w_t$, of the latent state $x_t$.

First, following standard KF results, on each trial, the participant is assumed to hold a prior belief (indicated with (–) superscript) about the latent state, $x_t$ (*Equation 11*).

$$x_t | N_{1:t-1} \sim \mathcal{N}\left(m_t^{(-)}, w_t^{2(-)}\right) \tag{11}$$

On the first trial, before any observations, we set $m_1^{(-)} = E^0, w_1^{(-)} = w_0$ (free parameters). In light of the new observation, $N_t$ on trial $t$, the tracked mean and uncertainty of the latent state are reweighed based on the new evidence $N_t$ and its associated observation uncertainty $s$ as in *Equation 12*.

$$x_t | N_{1:t} \sim \mathcal{N}\left(\frac{s^2 m_t^{(-)} + w_t^{2(-)} N_t}{s^2 + w_t^{2(-)}}, \frac{s^2 w_t^{2(-)}}{s^2 + w_t^{2(-)}}\right) \tag{12}$$

We can then define the learning rate $\alpha_t$ (*Equation 13*),

$$\alpha_t = \frac{w_t^{2(-)}}{s^2 + w_t^{2(-)}} \tag{13}$$

to get the update rule for the new posterior beliefs (indicated with (+) superscript) about the mean (*Equation 14*) and uncertainty (*Equation 15*) of $x_t$.

$$m_t^{(+)} = m_t^{(-)}\left(1 - \alpha_t\right) + N_t \alpha_t \tag{14}$$

$$w_t^{2(+)} = w_t^{2(-)}\left(1 - \alpha_t\right) \tag{15}$$

Following this new belief, and the assumption about the environmental dynamics (volatility), the participant forms a new prior belief about the latent state $x_{t+1}$ for the next trial $t+1$ as in *Equation 16*.

$$x_{t+1} | N_{1:t} \sim \mathcal{N}\left(m_{t+1}^{(-)}, w_{t+1}^{2(-)}\right) \tag{16}$$

where

$$m_{t+1}^{(-)} = m_t^{(+)} \tag{17}$$

$$w_{t+1}^{2(-)} = w_t^{2(+)} + v^2 \tag{18}$$

We can simplify the notation to make it comparable to the RL models. We let $m_{t+1} = m_t^{(+)} = m_{t+1}^{(-)}$, and $w_{t+1}^2 = w_{t+1}^{2(-)} = w_t^{2(+)} + v^2$. Following a new observation at trial $t$, we calculate the prediction error (**Equation 19**) and learning rate (**Equation 20**).

$$\delta_t = y_t - m_t \tag{19}$$

$$\alpha_t = \frac{w_t^2}{w_t^2 + s^2} \tag{20}$$

We then update the belief about the mean (**Equation 21**) and uncertainty (**Equation 22**) of the latent state for the next trial.

$$\begin{aligned} m_{t+1} &= m_t \left(1 - \alpha_t\right) + N_t \alpha_t \\ &= m_t + \alpha_t \left(N_t - m_t\right) \end{aligned} \tag{21}$$

$$w_{t+1}^2 = w_t^2 \left(1 - \alpha_t\right) + v^2 \tag{22}$$

Now, mapping this onto the experiment, the mean of the latent state is participants' expectation $E_t = m_t$, and so we have participant perception rating modelled as in **Equation 23**.

$$\hat{P}_t \sim \mathcal{N}\left(P_t, \xi^2 exp\left\{C^{-1}\left(1 - c_t\right)\right\}^2\right) \tag{23}$$

and the prediction rating for the next trial as in **Equation 24**.

$$\hat{E}_{t+1} \sim \mathcal{N}\left(E_{t+1}, \xi^2 exp\left\{C^{-1}\left(1 - c_t\right)\right\}^2\right) \tag{24}$$

In total the model has six free parameters: $s$ (environmental stochasticity), $v$ (environmental volatility), $\xi$ (response noise), $E_0$ (initial belief about the mean), $w_0$ (initial belief about the uncertainty), and $C$ (confidence scaling factor).

### eKF

We can introduce the effect of expectation on the pain perception, by assuming that participants treat the thermal input as an imperfect indicator of the true level of pain (**Jepma et al., 2018**). In this case, the input, $N_t$, is modelled as in **Equation 25**:

$$N_t \sim \mathcal{N}\left(\pi_t, \epsilon^2\right) \tag{25}$$

which forms an expression for the likelihood of the observation and adds an additional level to the inference, slightly modifying the KF assumptions such that:

$$\pi_t \sim \mathcal{N}(x_t, s^2) \tag{26}$$

However, we can apply the standard KF results and Bayes' rule to arrive at simple update rules for the participants' belief about the mean and uncertainty of the latent state $x_t$. From this, we get a prior on the $\pi_t$ defined in **Equation 27**:

$$\pi_t | N_{1:\,t-1} \sim \mathcal{N}\left(m_t^{(-)}, w_t^{2(-)} + s^2\right) \tag{27}$$

which, following a new input $N_t$, gives us the posterior belief about $\pi_t$ as in **Equation 28**.

$$\pi_t | N_{1:\,t} \sim \mathcal{N}\left(\frac{\epsilon^2 m_t^{(-)} + \left(s^2 + w_t^{2(-)}\right) N_t}{\epsilon^2 + s^2 + w_t^{2(-)}}, \frac{\epsilon^2 \left(s^2 + w_t^{2(-)}\right)}{\epsilon^2 + s^2 + w_t^{2(-)}}\right) \tag{28}$$

Now, if we define $\gamma_t$ as in **Equation 29**:

$$\gamma_t = \frac{\epsilon^2}{\epsilon^2 + s^2 + w_t^{2(-)}} \tag{29}$$

We have that the posterior belief about the mean level of pain $\pi_t$ is calculated as:

$$P_t^{(+)} = \gamma_t m_t^{(-)} + \left(1 - \gamma_t\right) N_t \tag{30}$$

which is a weighted sum of the input $N_t$ and participant expectation about the latent state $x_t$, governed by the perceptual weight $\gamma_t$, analogously to the eRL model. Finally, the posterior belief about $x_t$ is obtained in *Equation 31*.

$$x_t | N_{1:t} \sim \mathcal{N}\left(\frac{\left(\epsilon^2 + s^2\right) m_t^{(-)} + w_t^{2(-)} N_t}{\epsilon^2 + s^2 + w_t^{2(-)}}, \frac{\left(\epsilon^2 + s^2\right) w_t^{2(-)}}{\epsilon^2 + s^2 + w_t^{2(-)}}\right) \tag{31}$$

Now, setting the learning rate as in *Equation 32*:

$$\alpha_t = \frac{w_t^2}{\epsilon^2 + w_t^2 + s^2} \tag{32}$$

we get:

$$m_t^{(+)} = m_t^{(-)} \left(1 - \alpha_t\right) + N_t \alpha_t \tag{33}$$

$$w_t^{2(+)} = w_t^{2(-)} \left(1 - \alpha_t\right) \tag{34}$$

Next, following the same notation simplification as before, we get the update rules for the prior belief about the mean (*Equation 35*) and uncertainty (*Equation 36*) of the latent state $x_{t+1}$ for the next trial.

$$\begin{aligned} m_{t+1} &= m_t \left(1 - \alpha_t\right) + N_t \alpha_t \\ &= m_t + \alpha_t \left(N_t - m_t\right) \end{aligned} \tag{35}$$

$$w_{t+1}^2 = w_t^2 \left(1 - \alpha_t\right) + v^2 \tag{36}$$

as well as the expression for subjective perception, $P_t$, at trial $t$ (*Equation 37*).

$$P_t = \gamma_t m_t + \left(1 - \gamma_t\right) N_t \tag{37}$$

The perception and prediction responses are modelled analogously as the KF model. In total, the model has seven free parameters: $\epsilon$ (subjective noise), $s$ (environmental stochasticity), $v$ (environmental volatility), $\xi$ (response noise), $E_0$ (initial belief about the mean), $w_0$ (initial belief about the uncertainty), and $C$ (confidence scaling factor).

### Random model

As a baseline, we also included a model that performs a random guess. The perceptual/prediction ratings were modelled as in *Equation 38*.

$$\hat{P}_t \sim \mathcal{N}\left(R, \xi^2 exp\left\{C^{-1} \left(1 - c_t\right)\right\}^2\right) \tag{38}$$

The model has three free parameters: $R$, $\xi$, and $C$, where $R$ is a constant value that participants respond with.

## Model fitting

Model parameters were estimated using hierarchical Bayesian methods, performed with RStan package (v. 2.21.0) (*Stan Development Team, 2019*) in R (v. 4.0.2) based on Markov Chain Monte Carlo techniques (No-U-Turn Hamiltonian Monte Carlo). For the individual-level parameters we used non-centred parametrisation (*Papaspiliopoulos et al., 2007*). For the group-level parameters we used

$\mathcal{N}(0, 1)$ priors for the mean, and the gamma-mixture representation of the Student's-t(3,0,1) for the scale (***Stan Development, 2022***). Parameters in the (0, 1) range were constrained using Phi_approx - a logistic approximation to the cumulative Normal distribution (***Bowling et al., 2009***).

For each condition and each of the four chains, we ran 6000 samples (after discarding 6000 warm-up ones). For each condition, we examined R-hat values for each individual- (including the $\mathcal{N}(0, 1)$ error term from the non-centred parametrisation) and group-level parameters from each model to verify whether the Markov chains have converged. At the group-level and individual-level, all R-hat values had a value <1.1, indicating convergence. In the random response model, 0.01–0.16% iterations saturated the maximum tree depth of 11.

## Model comparison

For model comparison, we used R package loo, which provides efficient approximate leave-one-out (LOO) cross-validation. The package allows to estimate the difference in models' expected predictive accuracy through the difference in ELPD (***Vehtari et al., 2017***). By looking at the ratio between the ELPD difference and the SE of the difference, we get the sigma effect - a heuristic for significance of such model differences. There's no agreed-upon threshold of SEs that determines significance, but the higher the sigma difference, the more robust is the effect. The closeness of fit can also be captured with LOO information criterion (LOOIC), where the lower LOOIC values indicate better fit.

## Parameter comparison

For the comparison of group-level parameters between conditions, we extracted 95% high-density intervals of the permuted and merged (across chains) posterior samples of each group-level parameter (***Kruschke, 2023***). To assess significant differences between conditions, we calculated a difference between such defined intervals. In the Bayesian scenario, a significant difference is indicated by the interval not containing the value 0 (***Aylward et al., 2019***; ***Ahn et al., 2017***).

## Parameter and model recovery

To asses the reliability of our modelling analysis (***Wilson and Collins, 2019***), for each model we performed parameter recovery analysis, where we simulated participants' responses using newly drawn individual-level parameters from the group-level distributions.

We repurposed existing sequences of noxious inputs in the [1, 13] range (pre-transformation). When then applied a linear transformation to the input sequences using sampled slope and intercept coefficients from a Gaussian distribution of these coefficients that we estimated based on our dataset using R's fitdistrplus package. Furthermore, we simulated the confidence ratings based on lag-1 auto-correlation across a moving window of the transformed input sequence.

We then fit the same model to the simulated data and calculated Pearson correlation coefficients $r$ between the generated and estimated individual-level parameters. The higher the coefficient $r$, the more reliable the estimates are, which can be categorised as: poor (if $r<0.5$); fair (if $0.5<r<0.75$); good ($0.75<r<0.9$); excellent (if $r>0.9$) (***White et al., 2018***). Results are reported in ***Appendix 1—table 3*** and ***Appendix 1—figures 6–11***.

We also performed model recovery analysis (***Wilson and Collins, 2019***), where we first simulated responses using each model and then fit each model-specific dataset with each model. We then counted the number of times a model fit the simulated data best (according to the LOOIC rule), effectively creating an $M{\times}M$ confusion matrix, where $M$ is the number of models. In the case where we have a diagonal matrix of ones, the models are perfectly recoverable and hence as reliable as possible. Results are reported in ***Appendix 1—table 4***.

In ***Appendix 1—Tables 6–9*** we report bulk and tail effective sample size (ESS) for each condition, for each model and parameter.

## Acknowledgements

The study was funded by an MRC Career Development Award to FM (MR/T010614/1) and a UKRI Advanced Pain Discovery Platform grant to both FM and BS (MR/W027593/1). BS was also funded by Wellcome (214251/Z/18/Z), Versus Arthritis (21537), and IITP (MSIT 2019-0-01371). This work has been performed using resources provided by the Cambridge Tier-2 system operated by the University

of Cambridge Research Computing Service (https://www.hpc.cam.ac.uk/) funded by EPSRC Tier-2 capital grant (EP/T022159/1). HPC access was additionally funded by an EPSRC research infrastructure grant to FM. We are grateful to Prof. Máté Lengyel and Prof. Deborah Talmi for helpful discussions about the study. For the purpose of open access, the author has applied a Creative Commons Attribution (CC BY) licence to any Author Accepted Manuscript version arising from this submission.

## Additional information

### Funding

| Funder | Grant reference number | Author |
| --- | --- | --- |
| Medical Research Council | MR/T010614/1 | Flavia Mancini |
| Medical Research Council | MR/W027593/1 | Ben Seymour Flavia Mancini |
| Wellcome Trust | 10.35802/214251 | Ben Seymour |
| Versus Arthritis | 21537 | Ben Seymour |
| IITP | MSIT 2019-0-01371 | Ben Seymour |
| University of Cambridge Research Computing Service | EP/T022159/1 | Flavia Mancini |
| NIHR Oxford Health BRC | NIHR203316 | Ben Seymour |

The funders had no role in study design, data collection and interpretation, or the decision to submit the work for publication. The views expressed are those of the author(s) and not necessarily those of the NIHR or the Department of Health and Social Care or other funders. For the purpose of Open Access, the authors have applied a CC BY public copyright license to any Author Accepted Manuscript version arising from this submission.

### Author contributions

Jakub Onysk, Conceptualization, Formal analysis, Investigation, Visualization, Methodology, Writing - original draft, Project administration, Writing - review and editing; Nicholas Gregory, Conceptualization, Formal analysis, Investigation, Writing - review and editing; Mia Whitefield, Formal analysis, Methodology; Maeghal Jain, Formal analysis; Georgia Turner, Conceptualization, Methodology; Ben Seymour, Writing - review and editing; Flavia Mancini, Conceptualization, Supervision, Funding acquisition, Methodology, Writing - original draft, Writing - review and editing

### Author ORCIDs

Jakub Onysk ⓘ http://orcid.org/0000-0003-0876-5465
Ben Seymour ⓘ http://orcid.org/0000-0003-1724-5832
Flavia Mancini ⓘ http://orcid.org/0000-0001-8441-9236

### Ethics

All participants gave informed written consent to take part in the study, which was approved by the the ethics committee of the Department of Engineering, University of Cambridge.

Reviewer #1 (Public Review): https://doi.org/10.7554/eLife.90634.3.sa1
Reviewer #3 (Public Review): https://doi.org/10.7554/eLife.90634.3.sa2
Author response https://doi.org/10.7554/eLife.90634.3.sa3

## Additional files

### Supplementary files
• MDAR checklist

## Data availability

All code and data are openly available on Zenodo (https://doi.org/10.5281/zenodo.11394627).

The following dataset was generated:

| Author(s) | Year | Dataset title | Dataset URL | Database and Identifier |
|---|---|---|---|---|
| Onysk J, Gregory N, Whitefield M, Jain M, Turner G, Seymour B, Mancini F | 2024 | Statistical learning shapes pain perception and prediction independently of external cues | https://doi.org/ 10.5281/zenodo. 11394627 | Zenodo, 10.5281/ zenodo.11394627 |

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

# Appendix 1

## Behavioural results

### Model-naive performance

For each sequence condition (volatility × stochasticity), we calculated the RMSE of participants' responses (type: perception and prediction ratings) and compared to the normative noxious input, as a measure of performance in the task. We analysed the RMSEs with a repeated measures ANOVA, with the results reported in *Appendix 1—table 1*.

Given the significant interaction between stochasticity and response type, we further ran a post hoc comparison tests for this effect, as reported in *Appendix 1—table 2*.

## Noxious inputs and responses

### Input transformation

We linearly transformed participants' responses to project them from the 1–13 range to 0–100 using a linear transformation we obtained from a regression of stimulus intensities onto pain ratings.

Plots of each participant's transformation can be found in *Appendix 1—figure 1*.

We superimposed participants' responses (perception and prediction ratings) onto the noxious input condition sequences in *Appendix 1—figure 2*. The black line marks the start of a new sequence condition.

Finally, we plotted participants' confidence ratings throughout the task in *Appendix 1—figure 3*.

## Model predictions

Following the model fitting procedure, in *Appendix 1—figure 4*, we plotted example model predicted ratings for both perception and prediction responses for each condition as compared with noxious input for one participant's responses.

Lastly, for each condition and for each participant we plotted model responses (perception and prediction) against participant responses in *Appendix 1—figure 5*. The grand mean correlation across participants for each condition and response type was calculated and included in the figure.

## Parameter recovery

The results of parameter recovery analysis for each parameter for each model are reported in *Appendix 1—table 3*. We recovered each individual (out of 27 participants) parameter ≈100 times and calculated the mean and SD of the correlation between the true and recovered parameter values.

Moreover, to assess the number of simulations needed, we calculated the average SD (and its error) of the correlation as a function of increasing number of simulation, as plotted in *Appendix 1—figure 6*. The average was obtained from the 500 randomly chosen permutations of different simulations at each $n$ (out of ≈100).

Next, we include scatter plots from the parameter recovery for each model and parameter in *Appendix 1—figures 7–11*.

## Model recovery

We also ran model recovery analysis as described in the Materials and methods. We report the confusion matrix of our analysis based on approximately 100 simulations (per model pair) in *Appendix 1—table 4*.

## Condition-wise model comparison

For each condition, we ran model comparison procedure as described in the Materials and methods. The results are reported in *Appendix 1—table 5*. In each condition, the expectation weighted models provided significantly better fit than models without this element.

## Model diagnostics

In *Appendix 1—tables 6–9*, we report bulk and tail ESS for each condition, for each model and parameter.

While some of the ESS values are below the recommended threshold of 100, indicating potential issues with parameter inference. This may be due to a low participant sample size, as well as small

number of trials per condition, hinting limited statistical power. Given that the Rhat values are all around 1, and that there are no divergent transitions, as well as a fairly good parameter recovery, we see this as a minor issue.

Lastly, in *Appendix 1—table 10* we model diagnostics for each condition, such as the estimated Bayesian fraction of missing information (E-BFMI), number of divergent transition, and E-BFMI values per chain.

## Modelling results

### Group-level differences between each condition

We plotted the estimate posterior distributions for each parameter of the model (including the across-trial average and the final learning rate and perceptual weighting term) in *Appendix 1—figure 12*. We found no group-level differences between conditions for any of the posterior distribution of the parameters in the winning eKF model.

### Individual-level differences between conditions

We estimated the individual-level parameters for each condition, and include their violin plots in *Appendix 1—figure 13*.

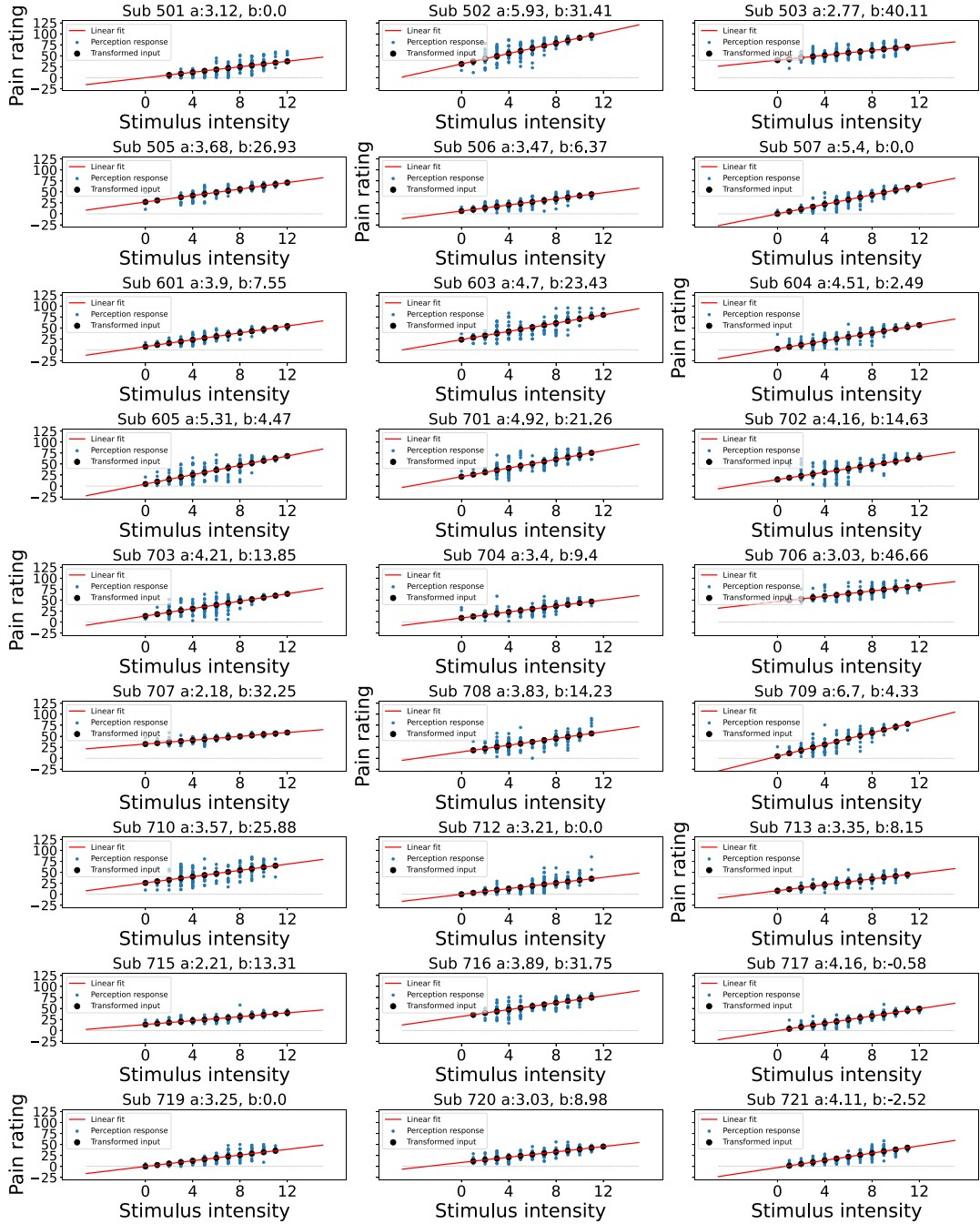

**Appendix 1—figure 1.** Linear transformation of the input at perception trials. Blue dots indicate participant's perception responses for a given level of stimulus intensity, black dots indicate transformed intensity values, a linear least squares regression was performed to achieve the best fitting line through participant's responses as shown in red, the intercept was constrained>0.

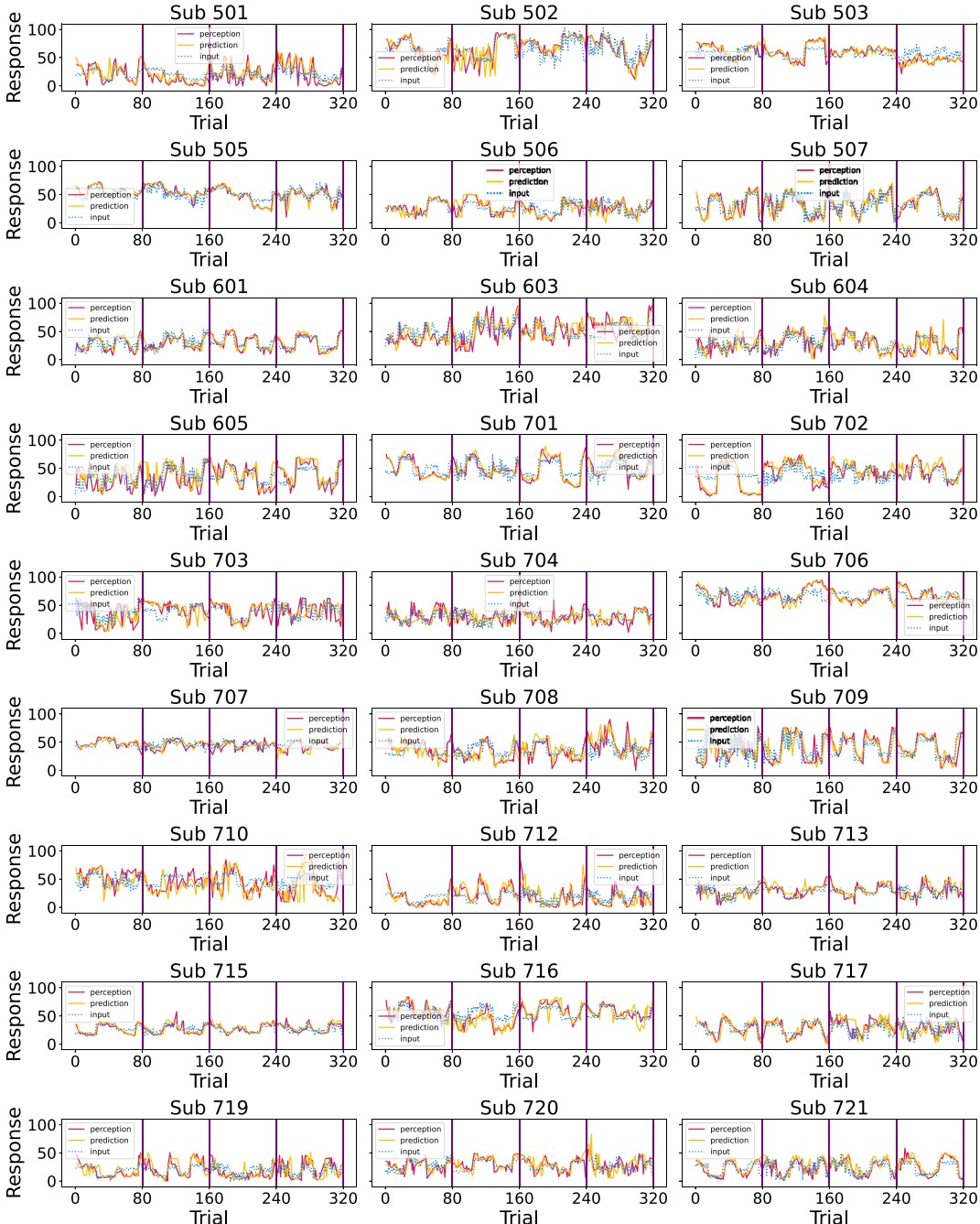

**Appendix 1—figure 2.** Participants' responses (red - perception; green - prediction) to the noxious input (dotted line) sequences. Vertical purple lines mark the end of each condition.

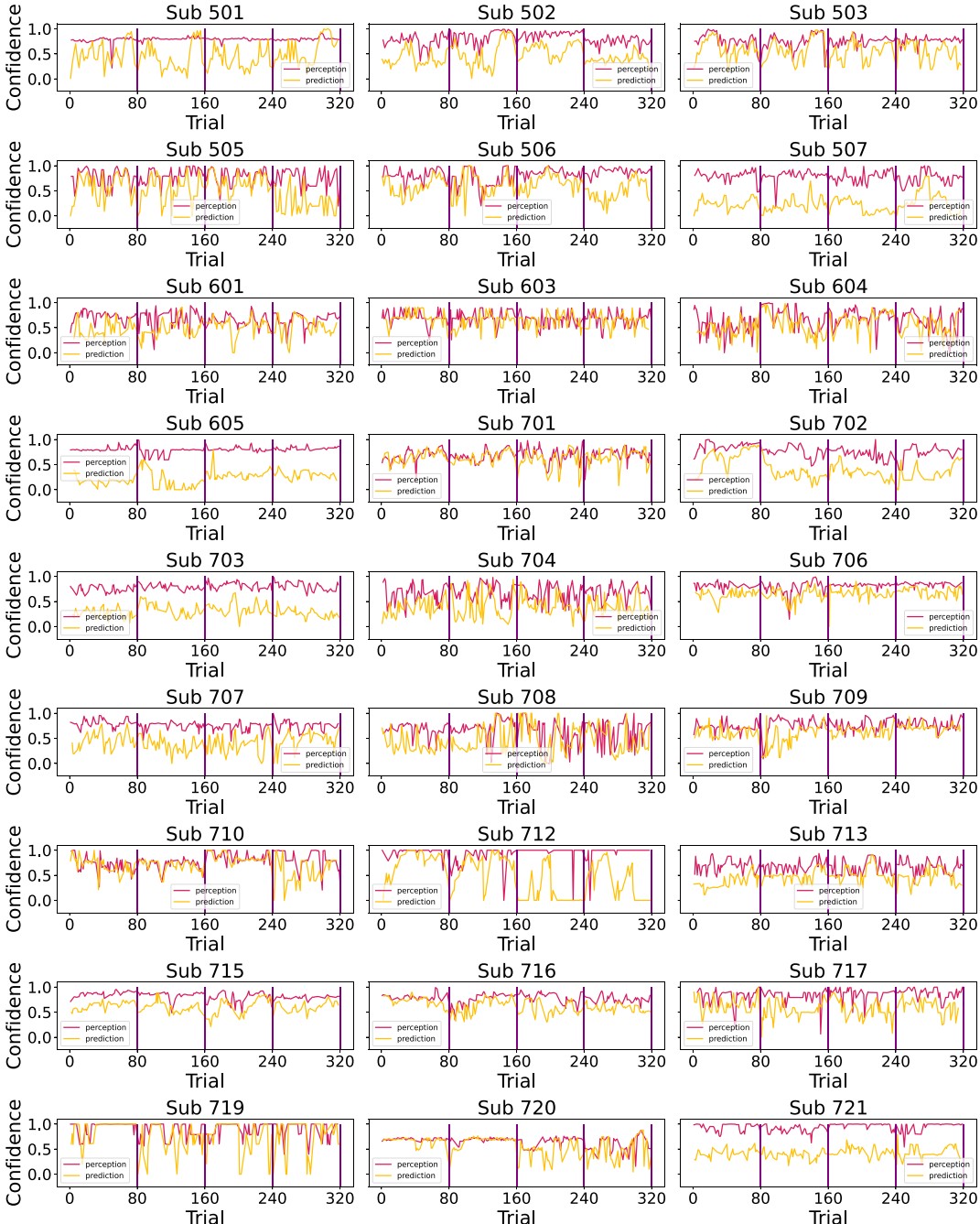

**Appendix 1—figure 3.** Participants' confidence ratings (red - perception; green - prediction) during the task. Vertical purple lines mark the end of each condition.

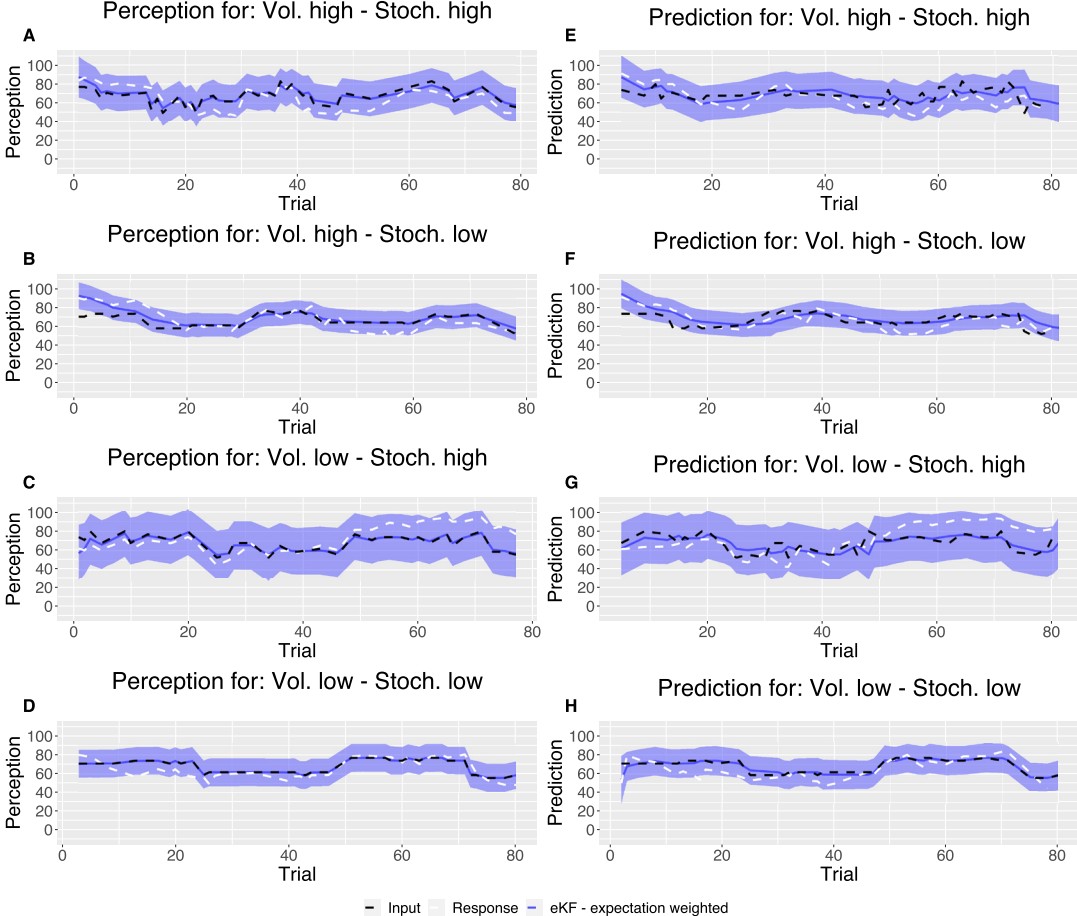

**Appendix 1—figure 4.** Example plot of the input sequences (black) for each condition, one participant's responses (white) and the winning, expectation weighted Kalman filter (eKF), model predictions (blue) including 95% confidence intervals (shaded blue) for (**A–D**) perception and (**E–H**) prediction.

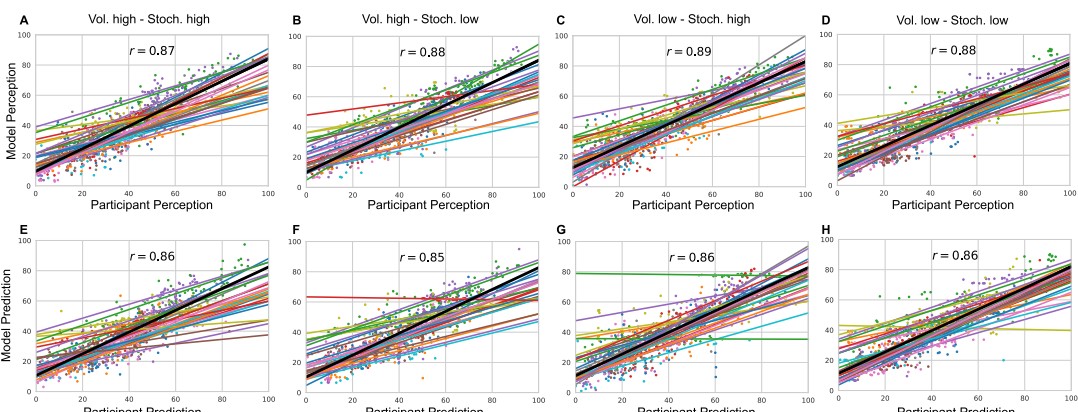

**Appendix 1—figure 5.** Model responses against participants' responses for each condition and each response type (**A–D**) perception and (**E–H**) prediction. The annotated value is the grand mean correlation across subjects for each condition and response type.

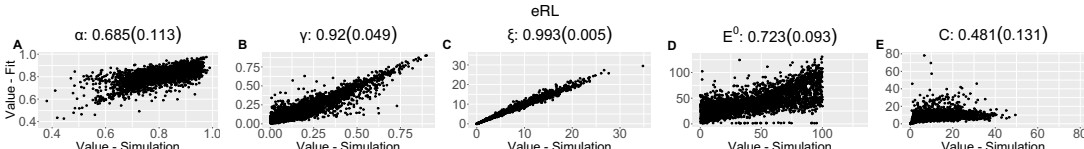

**Appendix 1—figure 6.** Parameter recovery average SD for: (**A**) eRL; (**B**) RL; (**C**) eKF; (**D**) KF; (**E**) Random model. The average SD is plotted as a function of simulation number averaged across 500 permutations of ≈100 simulations. The coloured shading corresponds to 1 SD around the average error.

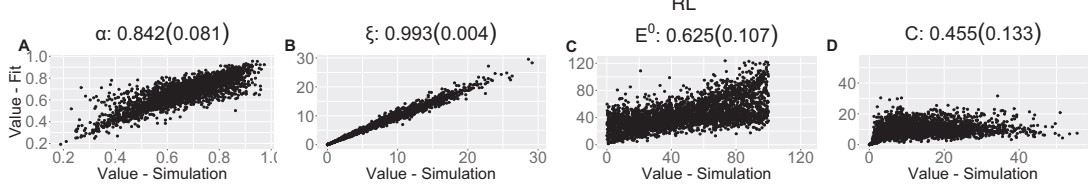

**Appendix 1—figure 7.** Parameter recovery scatter plot for expectation weighted reinforcement learning (eRL) model from ≈100 simulations for: (**A**) α; (**B**) γ; (**C**) ξ; (**D**) E0; (**E**) C parameter.

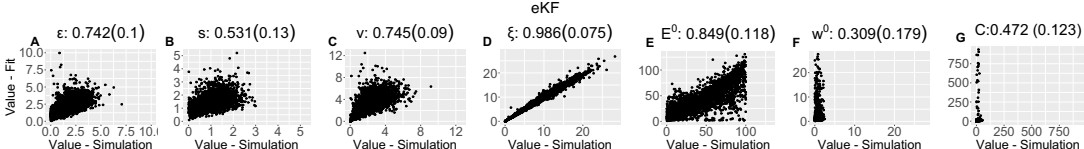

**Appendix 1—figure 8.** Parameter recovery scatter plot for reinforcement learning (RL) model from ≈100 simulations for: (**A**) α; (**B**) ξ; (**C**) E0; (**D**) C parameter.

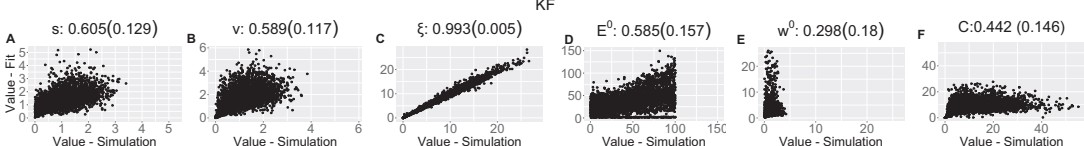

**Appendix 1—figure 9.** Parameter recovery scatter plot for expectation weighted Kalman filter (eKF) model from ≈100 simulations for: (**A**) ε; (**B**) s; (**C**) v; (**D**) ξ; (**E**) E0; (**F**) w0; (**G**) C parameter.

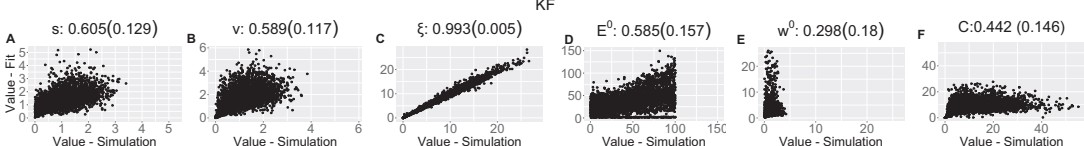

**Appendix 1—figure 10.** Parameter recovery scatter plot for Kalman filter (KF) model from ≈100 simulations for: (**A**) s; (**B**) v; (**C**) ξ; (**D**) E0; (**E**) w0; (**F**) C parameter.

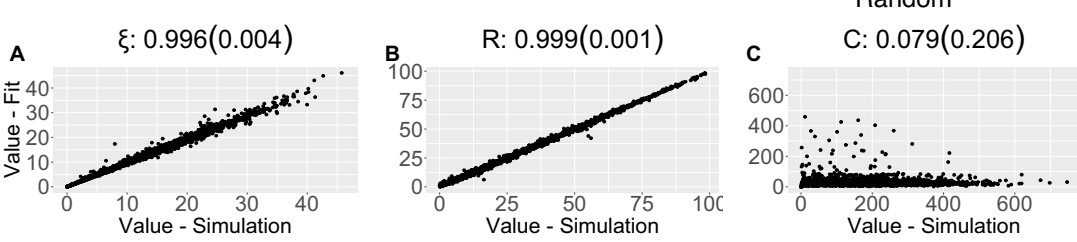

**Appendix 1—figure 11.** Parameter recovery scatter plot for random model from ≈100 simulations for: (**A**) ξ ; (**B**) R; (**C**) C parameter.

**Appendix 1—figure 12.** Group-level distributions for parameters for each condition for the expectation weighted Kalman filter (eKF) model.

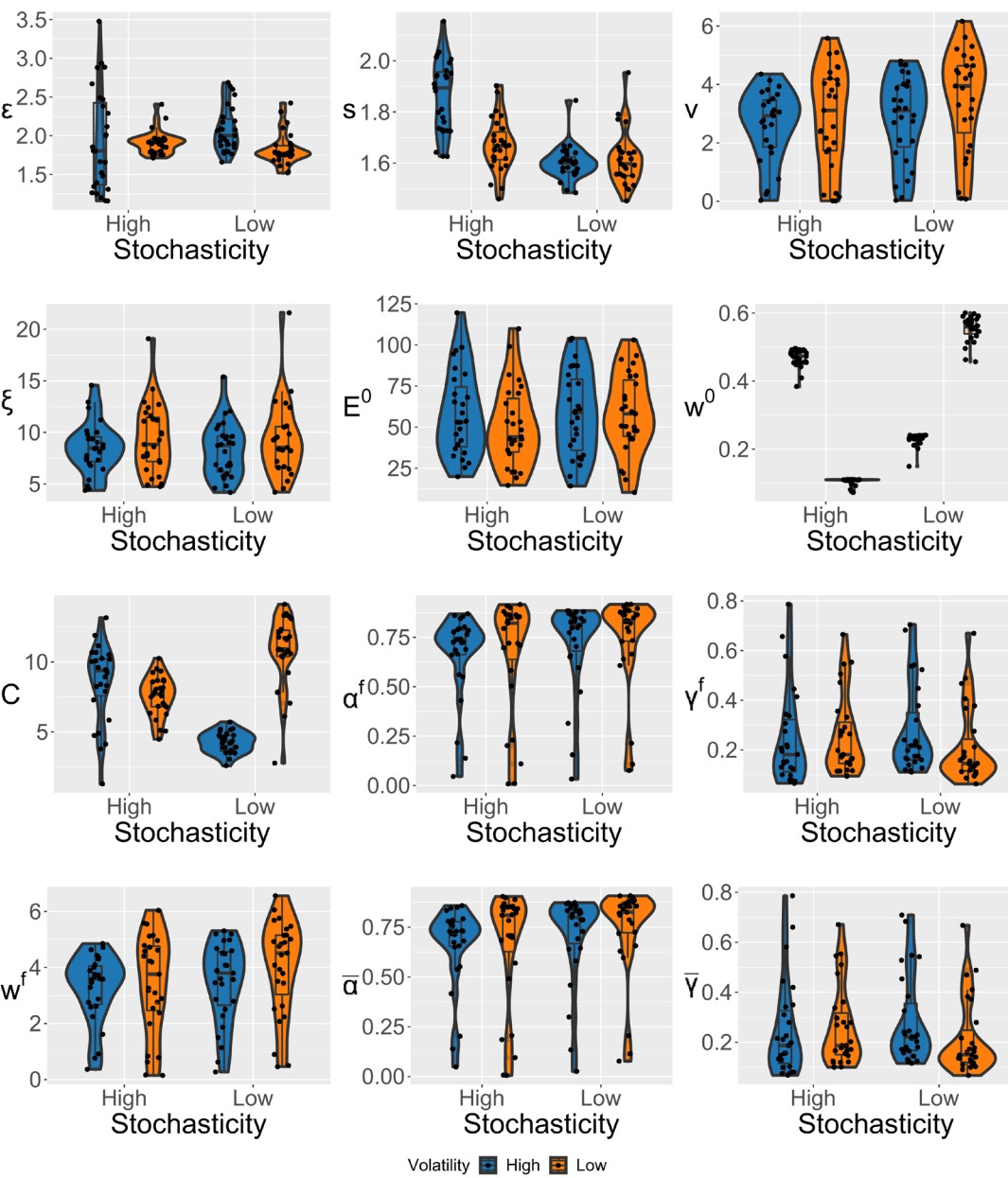

**Appendix 1—figure 13.** Violin plots (and box-plots) of individual-level parameters for each condition in the winning expectation weighted Kalman filter (eKF) model. Lower and upper hinges correspond to the first and third quartiles of partipants' errors (the upper/lower whisker extends from the hinge to the largest/smallest value no further than 1.5 * "Interquartile range" from the hinge); the line in the box corresponds to the median. Each condition has N=27 particpants.

**Appendix 1—table 1.** Within-subjects effects from repeated measures ANOVA of participant's RMSE scores with stochasticity, volatility, and response type factors.
SS - sum of squares, MS - mean square, RMSE - root mean square error

| Effect | SS | df | MS | $F$ | p | $\eta^2$ | $\eta_p^2$ |
|---|---|---|---|---|---|---|---|
| Volatility | 10.714 | 1 | 10.714 | 0.960 | 0.336 | 0.007 | 0.036 |
| Residuals | 290.166 | 26 | 11.160 | | | | |

*Appendix 1—table 1 Continued on next page*

*Appendix 1—table 1 Continued*

| Effect | SS | df | MS | F | p | $\eta^2$ | $\eta^2_p$ |
|---|---|---|---|---|---|---|---|
| Stochasticity | 113.964 | 1 | 113.964 | 19.939 | **<0.001*** | 0.074 | 0.434 |
| Residuals | 148.603 | 26 | 5.715 | | | | |
| Type | 365.000 | 1 | 365.000 | 85.109 | **<0.001*** | 0.237 | 0.766 |
| Residuals | 111.503 | 26 | 4.289 | | | | |
| Volatility × stochasticity | 0.006 | 1 | 0.006 | $5.688e^{-4}$ | 0.981 | $3.723e^{-6}$ | $2.188e^{-5}$ |
| Residuals | 261.912 | 26 | 10.074 | | | | |
| Volatility × type | 7.313 | 1 | 7.313 | 3.196 | 0.085 | 0.005 | 0.109 |
| Residuals | 59.487 | 26 | 2.288 | | | | |
| Stochasticity × type | 63.662 | 1 | 63.662 | 29.842 | **<0.001*** | 0.041 | 0.534 |
| Residuals | 55.466 | 26 | 2.133 | | | | |
| Volatility × stochasticity × type | 1.356 | 1 | 1.356 | 0.704 | 0.409 | $8.807e^{-4}$ | 0.026 |
| Residuals | 50.060 | 26 | 1.925 | | | | |

* indicates statistical significance at 0.05 level.

**Appendix 1—table 2.** Post hoc comparisons for the repeated measures ANOVA's interaction effect of stochasticity × type.

| | | | **95%CI for mean diff.** | | | | |
|---|---|---|---|---|---|---|---|
| | | Mean diff. | Lower | Upper | SE | t | $p_{bonf}$ |
| High, perception | Low, perception | 0.367 | −0.687 | 1.421 | 0.381 | 0.963 | 1.000 |
| | High, prediction | −3.686 | −4.636 | −2.735 | 0.345 | −10.688 | **<0.001*** |
| | Low, prediction | −1.147 | −2.329 | 0.034 | 0.430 | −2.665 | 0.062 |
| Low, perception | High, prediction | −4.053 | −5.234 | −2.871 | 0.430 | −9.415 | **<0.001*** |
| | Low, prediction | −1.514 | −2.464 | −0.564 | 0.345 | −4.390 | **<0.001*** |
| High, prediction | Low, prediction | 2.539 | 1.484 | 3.593 | 0.381 | 6.658 | **<0.001*** |

* indicates statistical significance at 0.05 level.

**Appendix 1—table 3.** Pearson correlation coefficient *r* (SD) from the parameter recovery analysis for each model.

**eRL**

| | $\alpha$ | $\gamma$ | $\xi$ | $E^0$ | $C$ | | |
|---|---|---|---|---|---|---|---|
| *r* (SD) | 0.685 (0.113) | 0.92 (0.049) | 0.993 (0.005) | 0.723 (0.093) | 0.481 (0.131) | | |

**RL**

| | $\alpha$ | $\xi$ | $E^0$ | $C$ | | | |
|---|---|---|---|---|---|---|---|
| *r* (SD) | 0.842 (0.081) | 0.993 (0.004) | 0.625 (0.107) | 0.455 (0.133) | | | |

**eKF**

| | $\epsilon$ | $s$ | $v$ | $\xi$ | $E^0$ | $w^0$ | $C$ |
|---|---|---|---|---|---|---|---|
| *r* (SD) | 0.742 (0.1) | 0.531 (0.13) | 0.745 (0.09) | 0.986 (0.075) | 0.849 (0.118) | 0.309 (0.179) | 0.472 (0.123) |

**KF**

| | $s$ | $v$ | $\xi$ | $E^0$ | $w^0$ | $C$ | |
|---|---|---|---|---|---|---|---|

*Appendix 1—table 3 Continued on next page*

*Appendix 1—table 3 Continued*

| | eRL | | | | | |
|---|---|---|---|---|---|---|
| *r* (SD) | 0.605 (0.129) | 0.589 (0.117) | 0.993 (0.005) | 0.585 (0.157) | 0.298 (0.18) | 0.442 (0.146) |
| | Random model | | | | | |
| | $\xi$ | *R* | *C* | | | |
| *r* (SD) | 0.996 (0.004) | 0.999 (0.001) | 0.079 (0.206) | | | |

**Appendix 1—table 4.** Confusion matrix from the model recovery analysis based on ≈100 simulations.

The *y*-axis indicates which model simulated the dataset, while the *x*-axis indicates which model fit the data based on leave-one-out information criterion (LOOIC).

| | | eRL | RL | eKF | KF | Random |
|---|---|---|---|---|---|---|
| | eRL | 0.327 | 0.173 | 0.404 | 0.096 | 0.000 |
| | RL | 0.223 | 0.234 | 0.223 | 0.319 | 0.000 |
| | eKF | 0.382 | 0.067 | 0.427 | 0.124 | 0.000 |
| | KF | 0.229 | 0.281 | 0.281 | 0.208 | 0.000 |
| | Random | 0.292 | 0.000 | 0.358 | 0.000 | 0.349 |
| Simulated | Fit | | | | | |

**Appendix 1—table 5.** Model comparison results for each condition.

| Condition | Model name | ELPD difference | SE difference | Sigma effect | LOOIC |
|---|---|---|---|---|---|
| | eKF - expectation weighted | 0.000 | 0.000 | | 15748.389 |
| | eRL - expectation weighted | −9.560 | 5.071 | 1.885 | 15767.509 |
| | RL | −139.407 | 61.362 | 2.272 | 16027.202 |
| | KF | −161.444 | 77.335 | 2.088 | 16071.277 |
| Vol. high Stoch. high | Random response | −730.600 | 77.009 | 9.487 | 17209.588 |
| | eKF - expectation weighted | 0.000 | 0.000 | | 15682.115 |
| | eRL - expectation weighted | −17.439 | 5.896 | 2.958 | 15716.993 |
| | RL | −131.817 | 35.936 | 3.668 | 15945.749 |
| | KF | −133.464 | 37.171 | 3.591 | 15949.042 |
| Vol. high Stoch. low | Random response | −824.346 | 79.148 | 10.415 | 17330.807 |
| | eKF - expectation weighted | 0.000 | 0.000 | | 15990.114 |
| | eRL - expectation weighted | −12.027 | 7.029 | 1.711 | 16014.169 |
| | RL | −149.338 | 43.874 | 3.404 | 16288.789 |
| | KF | −159.738 | 46.485 | 3.436 | 16309.590 |
| Vol. low Stoch. high | Random response | −831.096 | 84.549 | 9.830 | 17652.306 |
| | eKF - expectation weighted | 0.000 | 0.000 | | 15904.936 |
| | eRL - expectation weighted | −11.068 | 4.309 | 2.569 | 15927.072 |
| | RL | −70.588 | 16.643 | 4.241 | 16046.111 |
| | KF | −74.031 | 20.972 | 3.530 | 16052.997 |
| Vol. low Stoch. low | Random response | −901.792 | 107.244 | 8.409 | 17708.519 |

**Appendix 1—table 6.** Bulk and tail effective sample size (ESS) values for vol. high - stoch. high.

| Model | Param. | ESS (bulk) | ESS (tail) |
|---|---|---|---|
| | $\alpha$ | 58.166 | 47.491 |
| | $C$ | 90.5 | 79.142 |
| | $E^0$ | 54.655 | 137.729 |
| | $\xi$ | 31.233 | 47.726 |
| eRL | $\gamma$ | 39.509 | 49.335 |
| | $\alpha$ | 56.22 | 36.057 |
| | $C$ | 99.642 | 52.599 |
| | $E^0$ | 126.757 | 467.373 |
| RL | $\xi$ | 31.322 | 36.92 |
| | $C$ | 89.281 | 83.274 |
| | $E^0$ | 37.723 | 103.977 |
| | $\epsilon$ | 94.203 | 429.332 |
| | $v$ | 53.099 | 41.511 |
| | $s$ | 1665.566 | 4593.161 |
| | $w^0$ | 616458.467 | 467603.626 |
| eKF | $\xi$ | 31.322 | 47.1 |
| | $C$ | 101.584 | 55.345 |
| | $E^0$ | 122.76 | 512.134 |
| | $v$ | 114.644 | 54.015 |
| | $s$ | 438.028 | 730.579 |
| | $w^0$ | 904.643 | 6759.804 |
| KF | $\xi$ | 31.457 | 36.763 |
| | $R$ | 27.939 | 33.982 |
| | $C$ | 397.862 | 259.967 |
| Random | $\xi$ | 32.334 | 41.271 |

**Appendix 1—table 7.** Bulk and tail effective sample size (ESS) values for vol. high - stoch. low.

| Model | Param. | ESS (bulk) | ESS (tail) |
|---|---|---|---|
| | $\alpha$ | 86.32 | 60.849 |
| | $C$ | 235.396 | 373.736 |
| | $E^0$ | 43.489 | 109.903 |
| | $\xi$ | 30.471 | 36.664 |
| eRL | $\gamma$ | 42.125 | 55.178 |
| | $\alpha$ | 49.221 | 40.877 |
| | $C$ | 328.761 | 455.542 |
| | $E^0$ | 63.341 | 111.689 |
| RL | $\xi$ | 30.304 | 38.063 |

*Appendix 1—table 7 Continued on next page*

*Appendix 1—table 7 Continued*

| Model | Param. | ESS (bulk) | ESS (tail) |
|---|---|---|---|
| | $C$ | 227.813 | 363.944 |
| | $E^0$ | 33.393 | 104.395 |
| | $\epsilon$ | 376.691 | 1218.299 |
| | $v$ | 45.861 | 37.486 |
| | $s$ | 99526.69 | 148393.383 |
| | $w^0$ | 567627.288 | 634817.458 |
| eKF | $\xi$ | 30.438 | 36.66 |
| | $C$ | 328.005 | 448.632 |
| | $E^0$ | 57.467 | 124.471 |
| | $v$ | 293.426 | 480.255 |
| | $s$ | 164.454 | 598.211 |
| | $w^0$ | 412979.973 | 354163.251 |
| KF | $\xi$ | 30.16 | 38.105 |
| | $R$ | 28.397 | 32.922 |
| | $C$ | 1794.614 | 1170.459 |
| Random | $\xi$ | 30.204 | 34.896 |

**Appendix 1—table 8.** Bulk and tail effective sample size (ESS) values for vol. low - stoch. high.

| Model | Param. | ESS (bulk) | ESS (tail) |
|---|---|---|---|
| | $\alpha$ | 43.312 | 40.66 |
| | $C$ | 248.885 | 434.44 |
| | $E^0$ | 49.006 | 85.409 |
| | $\xi$ | 29.68 | 34.909 |
| eRL | $\gamma$ | 45.37 | 52.755 |
| | $\alpha$ | 39.911 | 35.351 |
| | $C$ | 433.949 | 435.575 |
| | $E^0$ | 181.442 | 618.317 |
| RL | $\xi$ | 29.527 | 36.192 |
| | $C$ | 248.848 | 418.003 |
| | $E^0$ | 35.363 | 51.728 |
| | $\epsilon$ | 1272.838 | 2427.211 |
| | $v$ | 41.144 | 40.915 |
| | $s$ | 2399.657 | 6854.212 |
| | $w^0$ | 612283.163 | 531588.25 |
| eKF | $\xi$ | 29.699 | 34.762 |

*Appendix 1—table 8 Continued on next page*

*Appendix 1—table 8 Continued*

| Model | Param. | ESS (bulk) | ESS (tail) |
|---|---|---|---|
| | $C$ | 423.339 | 417.747 |
| | $E^0$ | 88.749 | 302.863 |
| | $v$ | 58.795 | 47.015 |
| | $s$ | 206.969 | 672.666 |
| | $w^0$ | 499152.469 | 573964.793 |
| KF | $\xi$ | 29.511 | 36.341 |
| | $R$ | 27.892 | 32.919 |
| | $C$ | 269.239 | 106.139 |
| Random | $\xi$ | 29.69 | 44.38 |

**Appendix 1—table 9.** Bulk and tail effective sample size (ESS) values for vol. low - stoch. low.

| Model | Param. | ESS (bulk) | ESS (tail) |
|---|---|---|---|
| | $\alpha$ | 57.116 | 40.932 |
| | $C$ | 162.472 | 129.413 |
| | $E^0$ | 43.707 | 117.295 |
| | $\xi$ | 29.632 | 34.486 |
| eRL | $\gamma$ | 65.497 | 151.548 |
| | $\alpha$ | 45.892 | 37.244 |
| | $C$ | 158.681 | 98.898 |
| | $E^0$ | 80.406 | 441.719 |
| RL | $\xi$ | 29.558 | 35.077 |
| | $C$ | 149.16 | 126.209 |
| | $E^0$ | 38.88 | 73.732 |
| | $\epsilon$ | 653.635 | 1473.554 |
| | $v$ | 48.883 | 43.445 |
| | $s$ | 2263.547 | 9318.066 |
| | $w^0$ | 635517.969 | 313426.188 |
| eKF | $\xi$ | 29.699 | 34.721 |
| | $C$ | 158.729 | 105.929 |
| | $E^0$ | 71.438 | 457.431 |
| | $v$ | 91.988 | 69.957 |
| | $s$ | 287.835 | 895.249 |
| | $w^0$ | 527620.655 | 587092.529 |
| KF | $\xi$ | 29.527 | 35.147 |
| | $R$ | 28.474 | 38.123 |
| | $C$ | 2426.581 | 1279.66 |
| Random | $\xi$ | 29.532 | 34.731 |

**Appendix 1—table 10.** Model diagnostics for each condition - estimated Bayesian fraction of missing information (E-BFMI), number of divergent transition E-BFMI values per chain.

| Condition | Model | # chains low E-BFMI | # div. transitions | E-BFMI values |
|---|---|---|---|---|
| | eRL | 0 | 0 | 0.696 0.713 0.695 0.691 |
| | RL | 0 | 0 | 0.76 0.748 0.771 0.806 |
| | eKF | 0 | 0 | 0.755 0.767 0.771 0.759 |
| | KF | 0 | 0 | 0.633 0.596 0.547 0.563 |
| HVHS | Random | 0 | 0 | 0.842 0.851 0.843 0.835 |
| | eRL | 0 | 0 | 0.689 0.76 0.69 0.689 |
| | RL | 0 | 0 | 0.624 0.688 0.688 0.685 |
| | eKF | 0 | 0 | 0.741 0.764 0.753 0.779 |
| | KF | 0 | 0 | 0.654 0.734 0.689 0.674 |
| HVLS | Random | 0 | 0 | 0.883 0.779 0.836 0.833 |
| | eRL | 0 | 0 | 0.73 0.732 0.728 0.702 |
| | RL | 0 | 0 | 0.719 0.714 0.742 0.7 |
| | eKF | 0 | 0 | 0.753 0.755 0.792 0.766 |
| | KF | 0 | 0 | 0.75 0.768 0.729 0.754 |
| LVHS | Random | 0 | 0 | 0.864 0.849 0.883 0.845 |
| | eRL | 0 | 0 | 0.764 0.762 0.75 0.764 |
| | RL | 0 | 0 | 0.714 0.764 0.719 0.697 |
| | eKF | 0 | 0 | 0.783 0.751 0.772 0.77 |
| | KF | 0 | 0 | 0.705 0.695 0.702 0.726 |
| LVLS | Random | 0 | 0 | 0.835 0.829 0.852 0.847 |

