## [Editor Report · eLife assessment]

This study presents a **valuable** insight into a computational mechanism of pain perception. The evidence supporting the authors' claims is **compelling**. The work will be of interest to pain researchers working on computational models and cognitive mechanisms of pain in a Bayesian framework.

---

## [Referee Report · Reviewer #1 (Public Review)]

Summary:

This study examined the role of statistical learning in pain perception, suggesting that individuals' expectations about a sequence of events influence their perception of pain intensity. They incorporated the components of volatility and stochasticity into their experimental design and asked participants (n = 27) to rate the pain intensity, their prediction, and their confidence level. They compared two different inference strategies: Bayesian inference vs. heuristic-employing Kalman filters and model-free reinforcement learning. They showed that the expectation-weighted Kalman filter best explained the temporal pattern of participants' ratings. These results provide evidence for a Bayesian inference perspective on pain, supported by a computational model that elucidates the underlying process.

Strengths:

- Their experimental design included a wide range of input intensities and the levels of volatility and stochasticity. With elaborated computational models, they provide solid evidence that statistical learning shapes pain.

Weaknesses:

- Relevance to clinical pain: While the authors underscore the relevance of their findings to chronic pain, they did not include data pertaining to clinical pain.

---

## [Referee Report · Reviewer #3 (Public Review)]

The study investigated how statistical aspects of temperature sequences, such as manipulations of stochasticity (i.e., randomness of a sequence) and volatility (i.e., speed at which a sequence unfolded) influenced pain perception. Using an innovative stimulation paradigm and computational modelling of perceptual variables, this study demonstrated that perception is weighted by expectations. Overall, the findings support the conclusion that pain perception is mediated by expectations in a Bayesian manner. The provision of additional details during the review process strengthens the reliability of this conclusion. The methods presented offer tools and frameworks for further research in pain perception and can be extended to investigations into chronic pain processes.

---

## [Author Response]

The following is the authors’ response to the original reviews.

**eLife assessment**

This study presents a valuable insight into a computational mechanism of pain perception. The evidence supporting the authors’ claims is solid, although the inclusion of (1) more diverse candidate computational models, (2) more systematic analysis of the temporal regularity effects on the model fit, and (3) tests on clinical samples would have strengthened the study. The work will be of interest to pain researchers working on computational models and cognitive mechanisms of pain in a Bayesian framework.

Thank you very much again for considering the manuscript and judging it as a valuable contribution to understanding mechanisms of pain perception. We recognise the above-mentioned points of improvement and elaborate on them in the initial response to the reviewers.

Response to the reviewers

**Reviewer 1:**
Reviewer Comment 1.1 — Selection of candidate computational models: While the paper juxtaposes the simple model-free RL model against a Kalman Filter model in the context of pain perception, the rationale behind this choice remains ambiguous. It prompts the question: could other RL-based models, such as model-based RL or hierarchical RL, offer additional insights? A more detailed explanation of their computational model selection would provide greater clarity and depth to the study.

Initial reply: Thank you for this point. Our models were selected a-priori, following the modelling strategy from Jepma et al. (2018) and hence considered the same set of core models for clear extension of the analysis to our non-cue paradigm. The key question for us was whether expectations were used to weight the behavioural estimates, so our main interest was to compare expectation vs non-expectation weighted models.

Model-based and hierarchical RL are very broad terms that can be used to refer to many different models, and we are not clear about which specific models the reviewer is referring to. Our Bayesian models are generative models, i.e. they learn the generative statistics of the environment (which is characterised by inherent stochasticity and volatility) and hence operate model-based analyses of the stimulus dynamics. In our case, this happened hierarchically and it was combined with a simple RL rule.

Revised reply: We clarified our modelling choices in the ”Modelling strategy” subsection of the results section.

Reviewer Comment 1.2 — Effects of varying levels of volatility and stochasticity: The study commendably integrates varying levels of volatility and stochasticity into its experimental design. However, the depth of analysis concerning the effects of these variables on model fit appears shallow. A looming concern is whether the superior performance of the expectation-weighted Kalman Filter model might be a natural outcome of the experimental design. While the non-significant difference between eKF and eRL for the high stochasticity condition somewhat alleviates this concern, it raises another query: Would a more granular analysis of volatility and stochasticity effects reveal fine-grained model fit patterns?

Initial reply: We are sorry that the reviewer finds shallow ”the depth of analysis concerning the effects of these variables on model fit”. We are not sure which analysis the reviewer has in mind when suggesting a ”more granular analysis of volatility and stochasticity effects” to ”reveal fine-grained model fit patterns”. Therefore, we find it difficult to improve our manuscript in this regard. We are happy to add analyses to our paper but we would be greatful for some specific pointers. We have already provided:

• Analysis of model-naive performance across different levels of stochasticity and volatility (section 2.3, figure 3, supplementary information section 1.1 and tables S1-2)

• Model fitting for each stochasticity/volatility condition (section 2.4.1, figure 4, supplementary table S5)

• Group-level and individual-level differences of each model parameter across stochasticity/volatility conditions (supplementary information section 7, figures S4-S5).

• Effect of confidence on scaling factor for each stochasticity/volatility condition (figure 5)

Reviewer Comment 1.3 — Rating instruction: According to Fig. 1A, participants were prompted to rate their responses to the question, ”How much pain DID you just feel?” and to specify their confidence level regarding their pain. It is difficult for me to understand the meaning of confidence in this context, given that they were asked to report their *subjective* feelings. It might have been better to query participants about perceived stimulus intensity levels. This perspective is seemingly echoed in lines 100-101, ”the primary aim of the experiment was to determine whether the expectations participants hold about the sequence inform their perceptual beliefs about the intensity of the stimuli.”

Initial reply: Thank you for raising this question, which allows us to clarify our paradigm. On half of the trials, participants were asked to report the perceived intensity of the previous stimulus; on the remaining trials, participants were requested to predict the intensity of the next stimulus. Therefore, we did query ”participants about perceived stimulus intensity levels”, as described at lines 49-55, 296-303, and depicted in figure 1.

The confidence refers to the level of confidence that participants have regarding their rating - how sure they are. This is done in addition to their perceived stimulus intensity and it has been used in a large body of previous studies in any sensory modality.

Reviewer Comment 1.4 — Relevance to clinical pain: While the authors underscore the relevance of their findings to chronic pain, they did not include data pertaining to clinical pain. Notably, their initial preprint seemed to encompass data from a clinical sample (https://www.medrxiv.org /content/10.1101/2023.03.23.23287656v1), which, for reasons unexplained, has been omitted in the current version. Clarification on this discrepancy would be instrumental in discerning the true relevance of the study’s findings to clinical pain scenarios.

Initial reply: The preprint that the Reviewer is referring to was an older version of the manuscript in which we combined two different experiments, which were initially born as separate studies: the one that we submitted to eLife (done in the lab, with noxious stimuli in healthy participants) and an online study with a different statistical learning paradigm (without noxious stimuli, in chronic back pain participants). Unfortunately, the paradigms were different and not directly comparable. Indeed, following submission to a different journal, the manuscript was criticised for this reason. We therefore split the paper in two, and submitted the first study to eLife. We are now planning to perform the same lab-based experiment with noxious stimuli on chronic back pain participants. Progress on this front has been slowed down by the fact that I (Flavia Mancini) am on maternity leave, but it remains top priority once back to work.

Reviewer Comment 1.5 — Paper organization: The paper’s organization appears a little bit weird, possibly due to the removal of significant content from their initial preprint. Sections 2.12.2 and 2.4 seem more suitable for the Methods section, while 2.3 and 2.4.1 are the only parts that present results. In addition, enhancing clarity through graphical diagrams, especially for the experimental design and computational models, would be quite beneficial. A reference point could be Fig. 1 and Fig. 5 from Jepma et al. (2018), which similarly explored RL and KF models.

Initial reply: Thank you for these suggestions. We will consider restructuring the paper in the revised version.

Revised reply: We restructured introduction, results and parts of the methods. We followed the reviewer’s suggestion regarding enhancing clarity through graphical diagrams. We have visualised the experimental design in Figure 1D. Furthemore, we have visualised the two main computational models (eRL and eKF) in Figure 2, following from Jepma et al. (2018). As a result, we have updated the notation in Section 4.4 to be clearer and consistent with the graphical representation (rename the variable referring to observed thermal input from *Ot* to *Nt*).

Reviewer Comment 1.6 — In lines 99-100, the statement ”following the work by [23]” would be more helpful if it included a concise summary of the main concepts from the referenced work.- It would be helpful to have descriptions of the conditions that Figure 1C is elaborating on.- In line 364, the ”N {t}” in the sentence ”The observation on trial t, N {t}”, should be O {t}.

Initial reply: Thank you for spotting these and for providing the suggestions. We will include the correction in the revised version.

Revised reply: We have added the following regarding the lines 99-100:

”We build on the work by [23], who show that pain perception is strongly influenced by expectations as defined by a cue that predicts high or low pain. In contrast to the cue-paradigm from [23], the primary aim of our experiment was to determine whether the expectations participants hold about the sequence itself inform their perceptual beliefs about the intensity of the stimuli.”

See comment in the previous reply, regarding the notation change from *Ot* to *Nt*.

**Reviewer 2:**
Reviewer Comment 2.1 — This is a highly interesting and novel finding with potential implications for the understanding and treatment of chronic pain where pain regulation is deficient. The paradigm is clear, the analysis is state-of-the-art, the results are convincing, and the interpretation is adequate.

Initial reply: Thank you very much for these positive comments.

**Reviewer 3:**
Summary:I am pleased to have had the opportunity to review this manuscript, which investigated the role of statistical learning in the modulation of pain perception. In short, the study showed that statistical aspects of temperature sequences, with respect to specific manipulations of stochasticity (i.e., randomness of a sequence) and volatility (i.e., speed at which a sequence unfolded) influenced pain perception. Computational modelling of perceptual variables (i.e., multi-dimensional ratings of perceived or predicted stimuli) indicated that models of perception weighted by expectations were the best explanation for the data. My comments below are not intended to undermine or question the quality of this research. Rather, they are offered with the intention of enhancing what is already a significant contribution to the pain neuroscience field. Below, I highlight the strengths and weaknesses of the manuscript and offer suggestions for incorporating additional methodological details.Strengths:The manuscript is articulate, coherent, and skilfully written, making it accessible and engaging.- The innovative stimulation paradigm enables the exploration of expectancy effects on perception without depending on external cues, lending a unique angle to the research.- By including participants’ ratings of both perceptual aspects and their confidence in what they perceived or predicted, the study provides an additional layer of information to the understanding of perceptual decision-making. This information was thoughtfully incorporated into the modelling, enabling the investigation of how confidence influences learning.- The computational modelling techniques utilised here are methodologically robust. I commend the authors for their attention to model and parameter recovery, a facet often neglected in previous computational neuroscience studies.- The well-chosen citations not only reflect a clear grasp of the current research landscape but also contribute thoughtfully to ongoing discussions within the field of pain neuroscience.

Initial reply: We are really grateful for reviewer’s insightful comments and for providing useful guidance regarding our methodology. We are also thankful for highlighting the strengths of our manuscript. Below we respond to individual weakness mentioned in the reviews report.

Reviewer Comment 3.1 — In Figure 1, panel C, the authors illustrate the stimulation intensity, perceived intensity, and prediction intensity on the same scale, facilitating a more direct comparison. It appears that the stimulation intensity has been mathematically transformed to fit a scale from 0 to 100, aligning it with the intensity ratings corresponding to either past or future stimuli. Given that the pain threshold is specifically marked at 50 on this scale, one could logically infer that all ratings falling below this value should be deemed non-painful. However, I find myself uncertain about this interpretation, especially in relation to the term ”arbitrary units” used in the figure. I would greatly appreciate clarification on how to accurately interpret these units, as well as an explanation of the relationship between these values and the definition of pain threshold in this experiment.

Initial reply: Indeed, as detailed in the Methods section 4.3, the stimulation intensity was originally transformed from the 1-13 scale to 0-100 scale to match the scales in the participant response screens.

Following the method used to establish the pain threshold, we set the stimulus intensity of 7 as the threshold on the original 1-13 scale. However, during the rating part of the experiment, several of the participants never or very rarely selected a value above 50 (their individually defined pain threshold), despite previously indicating a moment during pain threshold procedure when a stimulus becomes painful. This then results in the re-scaled intensity values as well the perception rating, both on the same 0-100 scale of arbitrary units, to never go above the pain threshold. Please see all participant ratings and inputs in the Figure below. We see that it would be more illustrative to re-plot Figure 1 with a different exemplary participant, whose ratings go above the pain threshold, perhaps with an input intensity on the 1-13 scale on the additional right-hand-side y-axis. We will add this in the revised version as well as highlight the fact above.

Importantly, while values below 50 are deemed non-painful by participants, the thermal stimulation still activates C-fibres involved in nociception, and we would argue that the modelling framework and analysis still applies in this case.

Revised reply: We re-plotted Figure 1E-F with a different exemplary participant, whose rating go above the pain threshold. We also included all participant pain perception and prediction ratings, noxious input sequences and confidence ratings in the supplement in Figures S1-S3.

Reviewer Comment 3.2 — The method of generating fluctuations in stimulation temperatures, along with the handling of perceptual uncertainty in modelling, requires further elucidation. The current models appear to presume that participants perceive each stimulus accurately, introducing noise only at the response stage. This assumption may fail to capture the inherent uncertainty in the perception of each stimulus intensity, especially when differences in consecutive temperatures are as minimal as 1°C.

Initial reply: We agree with the reviewer that there are multiple sources of uncertainty involved in the process of rating the intensity of thermal stimuli - including the perception uncertainty. In order to include an account of inaccurate perception, one would have to consider different sources that contribute to this, which there may be many. In our approach, we consider one, which is captured in the expectation weighted model, more clearly exemplified in the expectation-weighted Kalman-Filter model (eKF). The model assumes participants perception of input as an imperfect indicator of the true level of pain. In this case, it turns out that perception is corrupted as a result of the expectation participants hold about the upcoming stimuli. The extent of this effect is partly governed by a subjective level of noise *ϵ*, which may also subsume other sources of uncertainty beyond the expectation effect. Moreover, the response noise *ξ*, could also subsume any other unexplained sources of noise.

**Author response image 1. sa3fig1:** Stimulis intensity transformation.

Revised reply: We clarified our modelling choices in the ”2.2 Modelling strategy” subsection.

Reviewer Comment 3.3 — A key conclusion drawn is that eKF is a better model than eRL. However, a closer examination of the results reveals that the two models behave very similarly, and it is not clear that they can be readily distinguished based on model recovery and model comparison results.

Initial reply: While, the eKF appears to rank higher than the eRL in terms of LOOIC and sigma effects, we don’t wish to make make sweeping statements regarding significance of differences between eRL and eKF, but merely point to the trend in the data. We shall make this clearer in the revised version of the manuscript. However, the most important result is that the models involving expectation-weighing are arguably better capturing the data.

Revised reply: We elaborated on the significance statements in the ”Modelling Results” subsection:

• We considered at least a 2 sigma effect as indication of a significant difference. In each condition, the expectation weighted models (eKF and eRL) provided better fit than models without this element (KF and RL; approx. 2-4 sigma difference, as reported in Figure 5A-D). This suggests that regardless of the levels of volatility and stochasticity, participants still weigh perception of the stimuli with their expectation.

and in the first paragraph of the Discussion:

• When varying different levels of inherent uncertainty in the sequences of stimuli (stochasticity and volatility), the expectation and confidence weighted models fitted the data better than models weighted for confidence but not for expectations (Figure 5A-D). The expectation-weighted bayesian (KF) model offered a better fit than the expectation-weighted, model-free RL model, although in conditions of high stochasticity this difference was short of significance. Overall, this suggests that participants’ expectations play a significant role in the perception of sequences of noxious stimuli.

We are aware of the limitations and lack of clear guidance regarding using sigma effects to establish significance (as per reviewer’s suggestion: see here). Here we decided to use the above-mentioned threshold of 2-sigma as an indication of significance, but note the potential limitations of the inferences - especially when distinguishing between eRL/eKF models.

Reviewer Comment 3.4 — Regarding model recovery, the distinction between the eKF and eRL models seems blurred. When the simulation is based on the eKF, there is no ability to distinguish whether either eKF or eRL is better. When the simulation is based on the eRL, the eRL appears to be the best model, but the difference with eKF is small. This raises a few more questions. What is the range of the parameters used for the simulations?

Initial reply: We agree that the distinction between eKF and eRL in the model recovery is not that clean-cut, which may in turn point to the similarity between the two models. To simulate the data for the model and parameter recovery analysis, we used the group means and variances estimated on the participant data to sample individual parameter values.

Reviewer Comment 3.5 — Is it possible that either eRL or eKF are best when different parameters are simulated? Additionally, increasing the number of simulations to at least 100 could provide more convincing model recovery results.

Initial reply: It could be a possibility, but would require further investigation and comparison of fits for different bins/ranges of parameters to see if there is any consistent advantage of one model over another is each bin. We will consider adding this analysis, and provide an additional 50 simulations to paint a more convincing picture.

Revised reply: We increased the number of simulations per model pair to ≈ 100 (after rejecting fits based on diagnostics criteria - E-BFMI and divergent transitions) and updated the confusion matrix (Table S4). Although the confusion between eRL and eKF remains, the model recovery shows good distinction between expectation weighted vs non-expectation weighted (and Random) models, which supports our main conclusion in the paper.

Reviewer Comment 3.6 — Regarding model comparison, the authors reported that ”the expectation-weighted KF model offered a better fit than the eRL, although in conditions of high stochasticity, this difference was short of significance against the eRL model.” This interpretation is based on a significance test that hinges on the ratio between the ELPD and the surrounding standard error (SE). Unfortunately, there’s no agreed-upon threshold of SEs that determines significance, but a general guideline is to consider ”several SEs,” with a higher number typically viewed as more robust. However, the text lacks clarity regarding the specific number of SEs applied in this test. At a cursory glance, it appears that the authors may have employed 2 SEs in their interpretation, while only depicting 1 SE in Figure 4.

Initial reply: Indeed, we considered 2 sigma effect as a threshold, however we recognise that there is no agreed-upon threshold, and shall make this and our interpretation clearer regarding the trend in the data, in the revision.

Revised reply: We clarify this further, as per our revised response to Comment 3.3 above. We have also added the following statement in section 4.5.1 (Methods, Model comparison): ”There’s no agreed-upon threshold of SEs that determines significance, but the higher the sigma difference, the more robust is the effect.”

Reviewer Comment 3.7 — With respect to parameter recovery, a few additional details could be included for completeness. Specifically, while the range of the learning rate is understandably confined between 0 and 1, the range of other simulated parameters, particularly those without clear boundaries, remains ambiguous. Including scatter plots with the simulated parameters on the xaxis and the recovered parameters on the y-axis would effectively convey this missing information.Furthermore, it would be beneficial for the authors to clarify whether the same priors were used for both the modelling results presented in the main paper and the parameter recovery presented in the supplementary material.

Initial reply: Thanks for this comment and for the suggestions. To simulate the data for the model and parameter recovery analysis, we used the group means and variances estimated on the participant data to sample individual parameter values. The priors on the group and individual-level parameters in the recovery analysis where the same as in the fitting procedure. We will include the requested scatter plots in the next iteration of the manuscript.

Revised reply: We included parameter recovery scatter plots for each model and parameter in the Supplement Figures S7-S11.

Reviewer Comment 3.8 — While the reliance on R-hat values for convergence in model fitting is standard, a more comprehensive assessment could include estimates of the effective sample size (bulk ESS and/or tail ESS) and the Estimated Bayesian Fraction of Missing Information (EBFMI), to show efficient sampling across the distribution. Consideration of divergences, if any, would further enhance the reliability of the results.

Initial reply: Thank you very much for this suggestion, we will aim to include these measures in the revised version.

Revised reply: We have considered the suggested diagnostics and include bulk and tail ESS values for each condition, model, parameter in the Supplement Tables S6-S9. We also report number of chain with low E-BFMI (0), number of divergent transitions (0) and the E-BFMI values per chain in Table S10.

Reviewer Comment 3.9 — The authors write: ”Going beyond conditioning paradigms based in cuing of pain outcomes, our findings offer a more accurate description of endogenous pain regulation.” Unfortunately, this statement isn’t substantiated by the results. The authors did not engage in a direct comparison between conditioning and sequence-based paradigms. Moreover, even if such a comparison had been made, it remains unclear what would constitute the gold standard for quantifying ”endogenous pain regulation.”

Initial reply: This is valid point, indeed we do not compare paradigms in our study, and will remove this statement in the future version.

Revised reply: We have removed this statement from the revised version.

Reviewer Comment 3.10 — In relation to the comment on model comparison in my public review, I believe the following link may provide further insight and clarify the basis for my observation. It discusses the use of standard error in model comparison and may be useful for the authors in addressing this particular point:see here

Initial reply: Thank you for this suggestion, we will consider the forum discussion in our manuscript.